# INVGC: Robust Cross-Modal Retrieval by Inverse Graph Convolution

**Xiangru Jian** and **Yimu Wang**[*]

University of Waterloo

{xiangru.jian,yimu.wang}@uwaterloo.ca

## Abstract

Over recent decades, significant advancements in cross-modal retrieval are mainly driven by breakthroughs in visual and linguistic modeling. However, a recent study shows that multi-modal data representations tend to cluster within a limited convex cone (as representation degeneration problem), which hinders retrieval performance due to the inseparability of these representations. In our study, we first empirically validate the presence of the representation degeneration problem across multiple cross-modal benchmarks and methods. Next, to address it, we introduce a novel method, called INVGC, a post-processing technique inspired by graph convolution and average pooling. Specifically, INVGC defines the graph topology within the datasets and then applies graph convolution in a subtractive manner. This method effectively separates representations by increasing the distances between data points. To improve the efficiency and effectiveness of INVGC, we propose an advanced graph topology, LOCALADJ, which only aims to increase the distances between each data point and its nearest neighbors. To understand why INVGC works, we present a detailed theoretical analysis, proving that the lower bound of recall will be improved after deploying INVGC. Extensive empirical results show that INVGC and INvGC w/LOCALADJ significantly mitigate the representation degeneration problem, thereby enhancing retrieval performance. Our code is available at link.

## 1 Introduction

Cross-modal retrieval (CMR) (Wang et al., 2021; Yu et al., 2023; Kim et al., 2023), which aims to enable flexible retrieval across different modalities, e.g., images, videos, audio, and text, has attracted significant research interest in the last few decades. The goal of CMR is to learn a pair of encoders that

---
[*]Corresponding author.

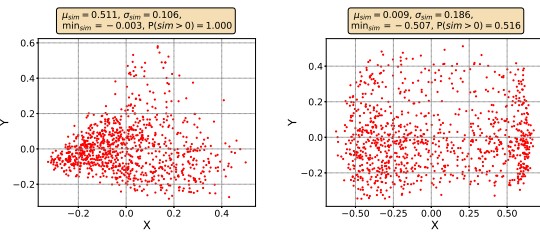

(a) Original representations. (b) Updated representations by INVGC.

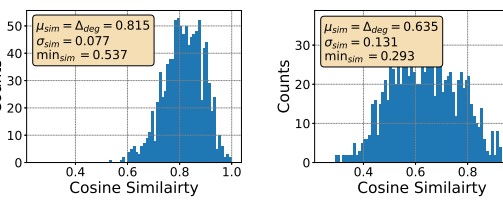

(c) Histogram of the similarity between nearest neighbors. (d) Histogram of the similarity between nearest neighbors after INVGC.

Figure 1: A 2D visualization of the representation of text data uniformly sampled from the MSCOCO dataset generated by CLIP (Radford et al., 2021) and then reduced by PCA. As shown in Figure 1a, the representation degeneration problem can be observed as all the representations are concentrated in a small convex cone, while after employing INVGC, representations are better separated. $\mu_{sim}$, $\sigma_{sim}$, and $\min_{sim}$ are the mean, standard deviation, and minimum of the similarity. $\Delta_{deg}$ is the score of representation degeneration problem defined in Equation (2) .

map data from different modalities into a common space where they can be directly compared. For example, in text-to-video retrieval, the objective is to rank gallery videos based on the features of the query text. Recently, inspired by the success in self-supervised learning (Radford et al., 2021), significant progress has been made in CMR, including image-text retrieval (Radford et al., 2021; Li et al., 2020; Wang et al., 2020a), video-text retrieval (Chen et al., 2020; Cheng et al., 2021; Gao et al., 2021; Lei et al., 2021; Ma et al., 2022; Park et al., 2022; Wang et al., 2022a,b; Zhao et al., 2022;

Wang and Shi, 2023; Wang et al., 2023), and audio-text retrieval (Oncescu et al., 2021), with satisfactory retrieval performances.

However, Liang et al. (2022) demonstrate that the current common learning paradigm of CMR leads to the representation degeneration problem, which concentrates all the representations in a small (convex) cone (Gao et al., 2019; Zhang et al., 2020) in image-text retrieval and the cosine similarity between any two points is positive, as shown in Figure 1a. Consequently, retrieval performance will be significantly affected (Radovanovic et al., 2010; Gao et al., 2019). Due to the limitation of space, detailed related works are presented in Appendix A.

In this paper, to step forward in addressing representation degeneration problem and further improve the retrieval performance, we first empirically test whether it is prevalent across various cross-modal retrieval models and datasets, including video, image, text, and audio. We found that the representations of MSCOCO generated by CLIP are gathered in a very narrow cone in the embedding space, proving the existence of representation degeneration problem, as shown in Figure 1a. This case does not stand alone since similar observations are observed across several cross-modal retrieval models and datasets as shown in Appendix C.3.

Next, to model how severe the representation degeneration problem is and the relationship between this problem and the retrieval performance, drawing inspiration from previous work (Gao et al., 2019; Zhang et al., 2020; Yu et al., 2022; Liang et al., 2022; Yang et al., 2022; Huang et al., 2021), we define it in CMR as the average similarity between each point and its nearest neighbor in the gallery set, as shown in Equation (2). We observe that the scores are very high across different datasets and methods. They are able to model this problem as a high score always leads to more concentrated data distribution as shown in Appendix C.3.

While CMR has suffered from this problem, on the other side, the graph convolution (Kipf and Welling, 2017) and average pooling (Boureau et al., 2010), which are widely employed in graph neural networks (Gilmer et al., 2017; Kipf and Welling, 2017; Velickovic et al., 2018) and deep neural networks (He et al., 2016; Yu et al., 2023), respectively, are designed to move the representations closer to each other if they are semantically similar (Baran-

wal et al., 2023). It might lead to the emergence of representation degeneration problem.

Drawing inspiration from the graph convolution and average pooling, we propose a novel method, INVGC, which separates representations by performing the graph convolution inversely to separate representations with a bigger margin as shown in Figure 1a. Specifically, different from the vanilla graph convolution, considering one modality, INVGC separates representations by subtracting the representation of the neighboring nodes from each node, instead of aggregating them as,

$$\mathbf{x_i}' = \mathbf{x_i} - r \sum_{j \neq i} S_{ij} \mathbf{x_j}, \forall i\,, \qquad (1)$$

where $\mathbf{x_i}'$ and $\mathbf{x_i}$ are the updated and the original representations, $S_{ij}$ is the similarity between the $i$-th and $j$-th data, and $r$ is a predefined hyper-parameter. As shown in Figure 1, INVGC better scatter representations and alleviates representation degeneration problem. Moreover, the histogram of similarity between any two points is more balanced with INVGC, as shown in Figure 1. To boost the effectiveness and efficiency of INVGC, we propose an advanced adjacency matrix LOCALADJ that directs INVGC w/LOCALADJ to focus on the nearest neighbors of each data point instead of considering all the data points.

To evaluate the effectiveness of INVGC and INVGC w/LOCALADJ, we conducted experiments on eight cross-modal benchmarks (Xu et al., 2016; Chen and Dolan, 2011; Fabian Caba Heilbron and Niebles, 2015; Hendricks et al., 2017; Lin et al., 2014; Plummer et al., 2017; Kim et al., 2019; Drossos et al., 2020). Experimental results show that INVGC alleviates representation degeneration problem across different datasets and methods and improves retrieval performance as a by-product.

In summary, our contributions are as follows[1]:

- We are the first to formalize the definition of representation degeneration in cross-modal retrieval and perform a theoretical analysis of the relationship between representation degeneration problem and retrieval performance.

- Inspired by the graph convolution, we propose the first post-processing method in cross-modal retrieval, namely INVGC, to alleviate representation degeneration problem without any training process or additional data.

---

[1]The code is released at link.

- We design an adjacency matrix, called LO-CALADJ, for the graph convolution, which leverages only the nearest neighbors of each data point instead of all the data. INVGC with LOCALADJ, namely INVGC w/LOCALADJ. It is shown to be more effective and efficient.

- Extensive experiments show that INVGC and INVGC w/LOCALADJ alleviate representation degeneration problem and improve retrieval performance as a by-product.

## 2 Preliminaries

### 2.1 Task Definition

In this paper, we focus on the representation degeneration problem in cross-modal retrieval (Wang et al., 2020b). Two modalities are denoted as $\mathcal{X}$ and $\mathcal{Y}$. $\mathcal{X}$ is the query modality, while $\mathcal{Y}$ is the gallery modality. The (test) gallery, denoted $G = \{\mathbf{g}_1, \ldots, \mathbf{g}_{N_g}\}$, contains all the representations of the (test) gallery data, where $N_g$ is the size. The query set is $Q = \{\mathbf{q}_1, \ldots, \mathbf{q}_{N_q}\}$, where $N_q$ is the number of queries. Usually, in cross-modal retrieval, the gallery data does not overlap with the training data. Additionally, as INVGC requires training (or validation) data to address the representation degeneration problem, we define the set of representations of training (or validation) query and gallery data as $\hat{Q} = \{\hat{\mathbf{q}}_1, \ldots, \hat{\mathbf{q}}_{N_{\hat{Q}}}\}$ and $\hat{G} = \{\hat{\mathbf{g}}_1, \ldots, \hat{\mathbf{g}}_{N_{\hat{G}}}\}$, respectively, where $N_{\hat{Q}}$ and $N_{\hat{G}}$ are the size of the training (or validation) query and gallery set, respectively. The similarity between two embeddings $\mathbf{a}$ and $\mathbf{b}$ is defined as $s_{\mathbf{a},\mathbf{b}} = \text{sim}(\mathbf{a}, \mathbf{b})$, where $\text{sim}(\cdot, \cdot)$ could be some measure of distance.

### 2.2 Representation Degeneration in Cross-modal Retrieval

Taking inspiration from Liang et al. (2022), we define the representation degeneration problem as the average similarity of pairs that are closest to each other,

$$\Delta_{deg} = \frac{1}{m} \sum_{\mathbf{x}} \text{sim}(\mathbf{x}, \mathbf{y}), \qquad (2)$$

where $\mathbf{y}$ is the closest data point to $\mathbf{x}$ while $m$ is the number of data points. A higher score, as shown in Appendix C.3, indicates a more severe representation degeneration issue, with each data point being closer to its nearest neighbors in the set.

To understand the relationship between the degeneration score (Equation (2)) and the retrieval performance, we present the following theorems.

**Theorem 1.** *Let $\mathbf{x}_1$ be any point in $G$, $\mathbf{x}_2$ be the nearest neighbor of $\mathbf{x}_1$ and $n$ be the dimension of the representation. Given a query point $\mathbf{q}$ that is semantically similar to $\mathbf{x}_1$ and sampled from $Q$, which follows an independent and identical uniform distribution in $\mathbb{R}^n$, the probability of a query point $\mathbf{q}$ to successfully retrieve $\mathbf{x_1}$, denoted $\text{P}(\mathbf{x_1}, b)$, is bounded by,*

$$\frac{n}{2} \cdot b^n > \text{P}(\mathbf{x_1}, b) > \frac{1}{4} \cdot b^{n+1},$$

*where $b = ||\mathbf{x_1} - \mathbf{x_2}||_2/2$.*

**Corollary 2.** *The ratio between the probability of successful retrieval of any two different neighborhood radii, namely $b_1$ and $b_2$, is*

$$\frac{\text{P}(\mathbf{x_1}, b_1)}{\text{P}(\mathbf{x_1}, b_2)} = \mathcal{O}\left(n \cdot \left(\frac{b_1}{b_2}\right)^n\right).$$

Due to space limitation, the proofs are deferred to Appendix D.1.

**Remark 1.** *These theorems show that a high similarity of the nearest neighbor, i.e., smaller $b$, leads to an exponentially lower probability for successful retrieval. Therefore, a higher $\Delta_{deg}$ score leads to bad retrieval performance.*

## 3 INVGC

To alleviate the representation degeneration problem and further boost the retrieval performance, we design a post-processing method, called INVGC, which does not need any additional training.

Our idea is generally based on the mathematical principles laid out in Equation (1) and Figure 1, where the inverse version of graph convolution is able to decrease the similarity between data points and their neighbors. For the sake of clear representation, we use the cosine similarity as the similarity metric in this section following Luo et al. (2022), as it is the most common practice in cross-modal retrieval[2]. **The formal definition of INVGC is presented in Section 3.3.**

### 3.1 Mathematical Intuition

INVGC originates from graph convolution, which is widely employed in graph neural networks (Kipf

---

[2]INVGC can be easily migrated to other similarity metrics adopted by different retrieval methods (Croitoru et al., 2021; Liu et al., 2019).

and Welling, 2017; Gilmer et al., 2017; Velickovic et al., 2018).

Specifically, graph convolution will concentrate all the embeddings of similar nodes which might lead to the concentration of similarity (de la Pena and Montgomery-Smith, 1995) and the data degeneration problem (Baranwal et al., 2023). On the other side, a similar operation, average pooling, has been employed in computer vision (He et al., 2016; Wang and Shi, 2023). Average pooling will aggregate the features that are location-based similar[3].

As a result, graph convolution and average pooling concentrate the representation of all the similar nodes and force them to become very similar to each other, potentially leading to representation degeneration problem. This observation inspires us to pose the following research question:

*Can the issue of representation degeneration problem in cross-modal retrieval be alleviated by conducting inverse graph convolution (*INVGC*)?*

To answer this question, we first give an inverse variant of graph convolution as follows,

$$\mathbf{x_i}' = \mathbf{x_i} - \sum_j A_{ij}\mathbf{x_j}, \qquad (3)$$

where $A$ is the adjacency matrix for the data in the gallery set. Since the adjacency matrix is not available, to encourage separating the nearest neighbor in terms of the similarity score of each node, we choose $A_{ij} = \text{sim}(i,j) := S_{ij}$ with detailed discussion in Section 3.2. Therefore, based on the inverse graph convolution, we propose the basic setting of INVGC as shown in Equation (1), which only considers one modality. We notice that it is able to alleviate the representation degeneration problem as shown in Figure 1.

Note that the ultimate goal of INVGC is to reduce $\Delta_{deg}$ score of the distribution of the representation of the gallery instead of merely the gallery set $G$, which can be regarded only as a sampled subset of the distribution with very limited size in practice. Therefore, the best approximation of the distribution is the training (or validation) gallery set $\hat{G}$ since it is the largest one we can obtain[4].

Similarly, the distribution of the query set $\hat{Q}$ is theoretically expected to be similar to that of $\hat{G}$ as

a basic assumption in machine learning (Bogolin et al., 2022). A detailed explanation of the claims is included in Appendix B.3. Moreover, as CMR needs to contrast the data points from both modalities, we utilize the (train or validation) gallery set $\hat{G}$ and the (train or validation) query set $\hat{Q}$ to better estimate the hidden distribution as shown in Equation (4),

$$\mathbf{x_i}' = \mathbf{x_i} - r_g \sum_{\mathbf{x_j} \in \hat{G}} S_{ij}^g \mathbf{x_j} - r_q \sum_{\mathbf{x_j} \in \hat{Q}} S_{ij}^q \mathbf{x_j}. \quad (4)$$

where $r_g$ and $r_q$ are two hyperparameters, $\mathcal{S}^g \in \mathbb{R}^{N_g \times N_{\hat{G}}}$ and $\mathcal{S}^q \in \mathbb{R}^{N_q \times N_{\hat{Q}}}$ is the adjacency matrices between every pair of embedding from $G$ and $\hat{G}$ and that from $Q$ and $\hat{Q}$, respectively.

To the best of our knowledge, INVGC is the first to utilize the (inverse) graph convolution for separating the representation of data. Instead of the commonly used capability of aggregation and message passing, we introduce an inverse variant of convolution that separates data representations compared to the vanilla graph convolution.

## 3.2 Constructing Adjacency matrix

Next, we need to establish the graph structure, i.e., build the adjacency matrix $\mathcal{S}^g$ and $\mathcal{S}^q$, since there is no natural graph structure in the dataset. The simplest idea will be that the edge weight between $i$ and $j$ equals 1, i.e., $S_{ij} = 1$, if the cosine similarity between $\mathbf{x_i}$ and $\mathbf{x_j}$, i.e., $\text{sim}(\mathbf{x_i}, \mathbf{x_j})$, is larger than 0 (or some thresholds). INVGC with this form of the adjacency matrix is an inverse variant of average pooling. It serves as a baseline in this study, denoted as AVGPOOL.

However, this scheme is not capable to reflect the degree of the similarity between $\mathbf{x_i}$ and $\mathbf{x_j}$ since it cannot precisely depict the relation between different data points.

As the magnitude of $S_{ij}$ directly controls how far will $\mathbf{x_i}$ go in the opposite direction of $\mathbf{x_j}$, a greedy design will be using the similarity score between them as the edge weight, i.e., $S_{ij} = \text{sim}(\mathbf{x_i}, \mathbf{x_j})$.

Therefore, we can calculate the adjacency matrix $\mathcal{S}^g$, which contains the similarity score between every pair of embedding from $G$ and $\hat{G}$, respectively. Specifically, the $(i,j)$-entry of $\mathcal{S}^g$ follows,

$$\mathcal{S}_{i,j}^g = \text{sim}(\mathbf{g_i}, \hat{\mathbf{g}}_{\mathbf{j}}). \qquad (5)$$

Similarly, the matrix $\mathcal{S}^q$ containing the similarity score between every pair of embedding from $G$ and

---

[3]The details of graph convolution and average pooling discussed in this study are deferred to the Appendices B.1 and B.2, respectively.

[4]In practice, the test queries are invisible to each other as the queries do not come at the same time. So the size of the query set $N_g$ is equal to 1.

$\hat{Q}$ is calculated as follows,

$$\mathcal{S}_{i,j}^q = \text{sim}(\mathbf{g_i}, \hat{\mathbf{q}_j}) . \qquad (6)$$

Now, with the well-defined $\mathcal{S}^q$ and $\mathcal{S}^g$, we can finally perform INVGC to alleviate representation degeneration problem.

As shown in Theorem 1, given any data point, the similarity of the nearest neighbor in the representation space is critical to retrieval performance. Inspired by this, when performing the inverse convolution on a node, we force INVGC to pay attention to those most similar nodes to it. This can be achieved by assigning edge weight only to the nodes having the top $k$ percent of the largest similarity scores relative to the given node. Specifically, each entry of $\mathcal{S}^g$ and $\mathcal{S}^q$ in this case is calculated as follows,

$$\mathcal{S}^g(i,j) = \begin{cases} \text{sim}(\mathbf{g_i}, \hat{\mathbf{g}_j}) & \text{, if } \text{sim}(\mathbf{g_i}, \hat{\mathbf{g}_j}) \geq P_i(\hat{G}, k) \\ 0 & \text{, else} \end{cases} \qquad (7)$$

$$\mathcal{S}^q(i,j) = \begin{cases} \text{sim}(\mathbf{q_i}, \hat{\mathbf{q}_j}) & \text{, if } \text{sim}(\mathbf{q_i}, \hat{\mathbf{q}_j}) \geq P_i(\hat{Q}, k) \\ 0 & \text{, else} \end{cases} \qquad (8)$$

where $P_i(\hat{G}, k)$ is the value of $k$-percentage largest similarity between node $i$ and all the nodes in $\hat{G}$. The same approach applies to $P_i(\hat{Q}, k)$ as well. We denote this refined adjacency matrix as LOCALADJ since it focuses on the local neighborhoods of each node.

### 3.3 Formal Definition of INVGC

With the well-formed adjacency matrices , we formally define INVGC to obtain the updated embedding of $G$, denoted as $G'$, in a matrix form as,

$$G' = \frac{1}{2}\left[\text{norm}(G - r_g \mathcal{S}^g \hat{G}) + \text{norm}(G - r_q \mathcal{S}^q \hat{Q})\right], \qquad (9)$$

where $\text{norm}(\cdot)$ normalizes a matrix with respect to rows, which is employed for uniformly distributing the intensity of convolution when cosine similarity is used and should be removed when adopting other similarity metrics and the adjacency matrices $\mathcal{S}^g$ and $\mathcal{S}^q$ can be calculated as Equations (5) to (7). Note that, compared to Equation (4), we separate the convolution on $\hat{G}$ and $\hat{Q}$ to pursue more robust results, for avoiding the distribution shifts between $\hat{G}$ and $\hat{Q}$ due to the imperfection of the representation method.

In summary, INVGC, as shown in Equation (9), is a brand new type of graph operation performed on the specifically designed graph structures (adjacency matrix) of data points in the cross-modal dataset. It helps alleviate the representation degeneration problem by separating data points located close to each other in the representation space.

After obtaining $G'$, the similarity between the $i$-th gallery points $\mathbf{g}'$ and a query point $\mathbf{q}$ is calculated as $sim(\mathbf{q}, \mathbf{g}')$.

## 4 Experiments

We conduct a series of experiments to demonstrate that INVGC can efficiently alleviate the representation degeneration problem in a post-hoc manner. We compare INVGC and INVGC w/LOCALADJ with the baseline performance produced by the representation model adopted. We also introduce the inverse version of average pooling, namely AVG-POOL, as another baseline. A series of ablation studies indicate that INVGC addresses the representation degeneration problem by reducing the similarity between points in the gallery set and is not sensitive to hyperparameters and the amount of training data.

### 4.1 Experimental and implementation settings

The implementation of INVGC exactly follows Equation (9) with the adjacency matrix defined in Equations (5) and (6) and a separate pair of tuned hyperparameters $r_g$ and $r_q$ for each retrieval model of each dataset. To balance the edge weight in $\mathcal{S}^q$ and $\mathcal{S}^g$ and make sure the scale of weights is stable, they subtract their average edge weights, respectively.

The only difference between INVGC w/LOCALADJ and INVGC is the adjacency matrix applied. Instead of the ones from Equations (5) and (6), we apply $\mathcal{S}^q$ and $\mathcal{S}^g$ defined in Equations (7) and (8). We choose the value of $k$ to be 1 (i.e., top 1% nearest neighbors) throughout the study while the proposed method is very robust to the value of the $k$. There is a large range of reasonable values that can boost the performance, proved by the ablation study in Section 4.3.

AVGPOOL is the simplest variant of inverse graph convolution with binary adjacency matrix where the edge weight between a pair of nodes is 1 if they are neighbors, or 0 otherwise. Then, following the approach in LOCALADJ, we update the embeddings using only the top $p$ percent of near-

Table 1: **Similarity measures within gallery set (MeanSim@1 equivalent to $\Delta_{deg}(G)$) on MSCOCO.** "MeanSim" and "MeanSim@k" refer to the mean of the similarity between all data points and that between all data points and their top-k neighbors.

| | Text-to-Video Retrieval | | | Video-to-Text Retrieval | | |
|---|---|---|---|---|---|---|
| | MeanSim ↓ | MeanSim@1↓ | MeanSim@10↓ | MeanSim ↓ | MeanSim@1↓ | MeanSim@10↓ |
| CLIP | 0.4717 | 0.8211 | 0.7803 | 0.5162 | 0.9029 | 0.8596 |
| CLIP w. INVGC | 0.4693 | 0.8142 | 0.7738 | 0.5137 | 0.8972 | 0.8542 |
| Difference to the baseline (%) | 0.51 ↓ | 0.84 ↓ | 0.83 ↓ | 0.48 ↓ | 0.63 ↓ | 0.63 ↓ |
| CLIP w. INVGC w/LOCALADJ | 0.4646 | 0.8059 | 0.7647 | 0.5105 | 0.8924 | 0.8477 |
| Difference to the baseline (%) | 1.51 ↓ | 1.85 ↓ | 2.00 ↓ | 1.09 ↓ | 1.16 ↓ | 1.38 ↓ |

Table 2: **Similarity measures between test gallery and query set on MSCOCO.** The nearest neighbor is considered with respect to the gallery set.

| | Text-to-Video Retrieval | | | Video-to-Text Retrieval | | |
|---|---|---|---|---|---|---|
| | MeanSim↓ | MeanSim@1↓ | MeanSim@10↓ | MeanSim↓ | MeanSim@1↓ | MeanSim@10↓ |
| CLIP | 0.1516 | 0.3282 | 0.3138 | 0.1516 | 0.3213 | 0.3009 |
| CLIP w. INVGC | 0.1500 | 0.3245 | 0.3101 | 0.1511 | 0.3198 | 0.2994 |
| Difference to the baseline(%) | 1.06↓ | 1.13↓ | 1.18↓ | 0.33 ↓ | 0.47↓ | 0.50 ↓ |
| CLIP w. INVGC w/LOCALADJ | 0.0635 | 0.2214 | 0.2035 | 0.0921 | 0.2414 | 0.2208 |
| Difference to the baseline(%) | 58.11 ↓ | 32.54 ↓ | 35.15 ↓ | 39.25 ↓ | 24.87 ↓ | 26.62 ↓ |

est neighbors. To provide a more comprehensive benchmark, we pick four values of $p$ from $10\%$ to $100\%$. Note that we also independently tune $r_g$ and $r_q$ for AVGPOOL to make sure the evaluation is fair. The comparison with AVGPOOL validates not only the effectiveness of INVGC but also the importance of the similarity-based adjacency matrix. The detailed setting is included in Appendix C.4.

## 4.2 Datasets and evaluation metrics

While we mainly focus our experiments on standard benchmarks for text-image retrieval, i.e., MSCOCO (Lin et al., 2014) and Flickr30k (Plummer et al., 2017), we also explore the generalization of INVGC by selecting four text-video retrieval benchmarks (MSR-VTT (Xu et al., 2016), MSVD (Chen and Dolan, 2011), Didemo (Hendricks et al., 2017), and ActivityNet (Fabian Caba Heilbron and Niebles, 2015)) and two text-audio retrieval benchmark (AudioCaps (Kim et al., 2019) and CLOTHO (Drossos et al., 2020)). The details of the benchmarks are deferred to Appendix C.1.

To evaluate the retrieval performance of INVGC, we use recall at Rank K (R@K, higher is better), median rank (MdR, lower is better), and mean rank (MnR, lower is better) as retrieval metrics, which are widely used in previous retrieval works (Luo et al., 2022; Ma et al., 2022; Radford et al., 2021).

## 4.3 INVGC and INVGC w/LOCALADJ

In this section, to answer a series of questions relating to INVGC and INVGC w/LOCALADJ, we investigate its performance on MSCOCO with CLIP given different settings. Due to space limitations, discussions of the sensitivity of INVGC w/LOCALADJ to $k$ and the complexity of both methods are included in the Appendix.

**RQ1: Is the data degeneration problem alleviated?** This problem can be firstly explained by the change of the similarity measure within the test gallery set $G$. As presented in Table 1, we collect the mean similarity of the gallery set of both tasks for three scenarios, the overall mean (MeanSim), the mean between the nearest neighbor(MeanSim@1), and the mean between nearest 10 neighbors (MeanSim@10). Note that Mean-Sim@1 is strictly equivalent to $\Delta_{deg}(G)$. It is very clear that INVGC and INVGC w/LOCALADJ do reduce the similarity on all accounts, especially MeanSim@1, indicating a targeted ability to alleviate the data degeneration issue.

Besides, given the assumption that the distribution of the test query set $Q$ is close to that of $G$, the similarity score of the gallery set to the query set for both tasks is also worth exploring. Since any point in $G$ should theoretically share an identical representation with its corresponding query in $Q$ and we try to maximize the similarity between them, we

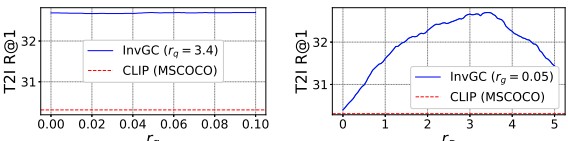

(a) Image-to-Text R@1 w.r.t (b) Image-to-Text R@1 w.r.t
$r_g$ $r_q$

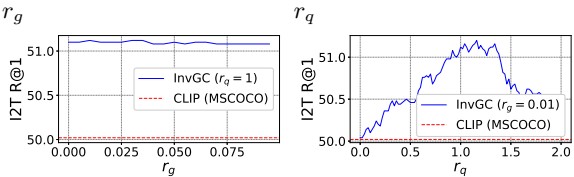

(c) Text-to-Image R@1 w.r.t (d) Text-to-Image R@1 w.r.t
$r_g$ $r_q$

Figure 2: Hyperpapameters sensitivity of INVGC on MSCOCO.

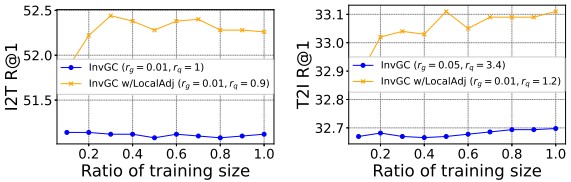

(a) Image-to-Text R@1 w.r.t (b) Text-to-Image R@1 w.r.t
the size of data we use.       the size of data we use.

Figure 3: Image-text retrieval performance w.r.t the size on MSCOCO.

exclude this similarity score between them. Consequently, we want the similarity to be as small as possible since it reflects the margin of the retrieval task, as a gallery point is supposed to be as far from the irrelevant queries as possible to have a more robust result. We adopt the same metrics as Table 1, and the results are presented in Table 2. Again, we observe a comprehensive decrease in the similarity score, especially between the nearest neighbors. Note that, compared to INVGC, INVGC w/LOCALADJ can better address the representation degeneration problem, validating that the design of LOCALADJ does help alleviate the problem by focusing more on the local information. Not for MSCOCO alone, we witness exactly similar results across all the datasets and methods, whose detail is included in **Continuation on RQ1** Appendix C.7.

**RQ2: Is INVGC (or INVGC w/LOCALADJ) sensitive to the hyperparameter $r_g$ and $r_q$?** To answer the question, we evaluate the R@1 metrics of INVGC with a very large range of hyperparameters respective to the optimal choice adopted. We defer the analysis of INVGC w/LOCALADJ to Appendix C.7. For each task, we fix one of $r_g$ or $r_q$

and tune the other to show the change on the R@1 metrics. The results of the MSCOCO dataset are shown in Figure 2. Although subject to some variation, INVGC constantly outperforms the baseline, which is presented as the red dashed line. This indicates that the proposed method can consistently improve performance with a very large range of parameters.

**RQ3: How much data is needed for both proposed methods?** Since we use the training query and gallery set as a sampled subset from the hidden distribution of representation, it is quite intuitive to ask if the performance of INVGC or INVGC w/LOCALADJ is sensitive to the size of this sampled subset. Therefore, we uniformly sample different ratios of data from both the training query and gallery set at the same time and evaluate the performance of both methods with the identical hyperparameters, as presented in Figure 3. From the results, we conclude that both proposed methods perform stably and robustly regardless of the size of the training data.

**RQ4: Is INVGC w/LOCALADJ sensitive to the hyperparameter $k$?** To address the question, we evaluate the R@1 metrics of INVGC w/LOCALADJ with a very large range of possible $k$ values (even in logarithmic scale) compared to the optimal choice adopted(i.e.,$k = 1$). For each task, we fix everything except the $k$ value.The results of MSCOCO dataset are shown in Figure 4. Compared with the baseline, it is safe to claim that the proposed INVGC w/LOCALADJ is very robust to the choice of $k$.

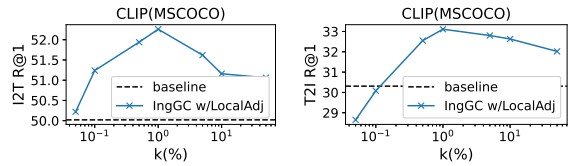

(a) Image-to-Text R@1 w.r.t (b) Text-to-Image R@1 w.r.t
the value of $k$ we use.       the value of $k$ we use.

Figure 4: The value of $k$ of INVGC w/LOCALADJ on MSCOCO.

### 4.4 Quantative results

In this section, we present the quantitative results of four cross-modal retrieval benchmarks. Across eight different benchmarks, INVGC and INVGC w/LOCALADJ significantly improve upon the baselines, demonstrating the superiority of our proposed methods.

Table 3: Retrieval performance on MSCOCO (5k split). Best in **Bold** and the second best is underlined.

| Normalization | Text-to-Image Retrieval | | | | |
| | R@1↑ | R@5↑ | R@10↑ | MdR↓ | MnR↓ |
|---|---|---|---|---|---|
| CLIP | 30.34 | 54.74 | 66.08 | 4.0 | 25.39 |
| +AvgPool(ratio=0.1) | 30.37 | 54.77 | 66.14 | 4.0 | 25.36 |
| +AvgPool(ratio=0.25) | 30.37 | 54.77 | 66.14 | 4.0 | 25.36 |
| +AvgPool(ratio=0.5) | 30.38 | 54.77 | 66.10 | 4.0 | 25.38 |
| +AvgPool(ratio=1) | 30.39 | 54.77 | 66.11 | 4.0 | 25.38 |
| +InvGC | 32.70 | 57.53 | 68.24 | 4.0 | **24.35** |
| +InvGC w/LocalAdj | **33.11** | 57.49 | 68.19 | 4.0 | 28.95 |
| Oscar | 52.50 | 80.03 | **87.96** | 1.0 | 10.68 |
| +AvgPool(ratio=0.1) | 52.52 | 80.04 | 87.95 | 1.0 | 10.70 |
| +AvgPool(ratio=0.25) | 52.52 | 80.03 | **87.96** | 1.0 | 10.68 |
| +AvgPool(ratio=0.5) | 52.51 | 80.00 | **87.96** | 1.0 | 10.67 |
| +AvgPool(ratio=1) | 52.50 | 80.02 | **87.96** | 1.0 | 10.68 |
| +InvGC | 52.63 | 80.05 | **87.96** | 1.0 | 10.72 |
| +InvGC w/LocalAdj | **52.93** | 80.05 | 87.78 | 1.0 | 11.09 |

Table 4: Retrieval performance on Flickr30k. Best in **Bold** and the second best is underlined.

| Normalization | Text-to-Image Retrieval | | | | |
| | R@1↑ | R@5↑ | R@10↑ | MdR↓ | MnR↓ |
|---|---|---|---|---|---|
| CLIP | 58.98 | 83.48 | 90.14 | 1.0 | 6.04 |
| +AvgPool(ratio=0.1) | 59.10 | 83.56 | 90.18 | 1.0 | 6.04 |
| +AvgPool(ratio=0.25) | 59.10 | 83.56 | 90.18 | 1.0 | 6.04 |
| +AvgPool(ratio=0.5) | 59.10 | 83.54 | 90.18 | 1.0 | 6.05 |
| +AvgPool(ratio=1) | 59.10 | 83.54 | 90.18 | 1.0 | 6.04 |
| +InvGC | 60.18 | 85.30 | **91.20** | 1.0 | **5.52** |
| +InvGC w/LocalAdj | **60.48** | 85.30 | 91.10 | 1.0 | 5.59 |
| Oscar | 71.60 | 91.50 | 94.96 | 1.0 | **4.24** |
| +AvgPool(ratio=0.1) | 71.62 | 91.44 | 94.92 | 1.0 | 4.25 |
| +AvgPool(ratio=0.25) | 71.66 | 91.50 | 94.94 | 1.0 | **4.24** |
| +AvgPool(ratio=0.5) | 71.66 | 91.50 | 94.92 | 1.0 | **4.24** |
| +AvgPool(ratio=1) | 71.62 | 91.52 | 94.96 | 1.0 | **4.24** |
| +InvGC | 71.68 | 91.46 | **95.06** | 1.0 | 4.25 |
| +InvGC w/LocalAdj | **71.74** | **91.56** | 94.98 | 1.0 | 4.29 |

Table 5: Retrieval performance on MSR-VTT (full split and 1k split). Best in **Bold** and the second best is underlined.

| Normalization | Text-to-Video Retrieval | | | | |
| | R@1 | R@5 | R@10 | MdR | MnR |
|---|---|---|---|---|---|
| CLIP4Clip | 44.10 | 71.70 | 81.40 | 2.0 | 15.51 |
| +AvgPool(ratio=0.1) | 44.20 | 71.60 | 81.50 | 2.0 | 15.55 |
| +AvgPool(ratio=0.25) | 44.20 | 71.60 | 81.50 | 2.0 | 15.55 |
| +AvgPool(ratio=0.5) | 44.20 | 71.50 | 81.50 | 2.0 | 15.54 |
| +AvgPool(ratio=1) | 44.10 | 71.60 | 81.50 | 2.0 | 15.52 |
| +InvGC | 44.40 | 71.90 | 81.60 | 2.0 | 15.36 |
| +InvGC w/LocalAdj | 44.40 | 71.70 | 81.20 | 2.0 | 15.65 |
| CLIP2Video | 46.00 | 71.60 | 81.60 | 2.0 | 14.51 |
| +AvgPool(ratio=0.1) | 45.90 | 71.70 | 81.50 | 2.0 | 14.52 |
| +AvgPool(ratio=0.25) | 46.10 | 71.80 | 81.50 | 2.0 | 14.53 |
| +AvgPool(ratio=0.5) | 46.00 | 71.70 | 81.50 | 2.0 | 14.53 |
| +AvgPool(ratio=1) | 45.90 | 71.70 | 81.50 | 2.0 | 14.52 |
| +InvGC | 46.20 | 71.70 | 81.30 | 2.0 | 14.44 |
| +InvGC w/LocalAdj | 46.60 | 72.10 | 81.70 | 2.0 | 14.50 |
| X-CLIP | 46.30 | 74.00 | 83.40 | 2.0 | 12.80 |
| +AvgPool(ratio=0.1) | 46.50 | 74.00 | 83.40 | 2.0 | 12.88 |
| +AvgPool(ratio=0.25) | 46.40 | 74.00 | 83.50 | 2.0 | 12.85 |
| +AvgPool(ratio=0.5) | 46.30 | 74.00 | 83.30 | 2.0 | 12.83 |
| +AvgPool(ratio=1) | 46.20 | 74.00 | 83.40 | 2.0 | 12.83 |
| +InvGC | 47.30 | 74.00 | 83.30 | 2.0 | 13.42 |
| +InvGC w/LocalAdj | 47.10 | 74.20 | 83.50 | 2.0 | 13.09 |

Table 6: Retrieval performance on ActivityNet. Best in **Bold** and the second best is underlined.

| Normalization | Text-to-Video Retrieval | | | | |
| | R@1 | R@5 | R@10 | MdR | MnR |
|---|---|---|---|---|---|
| CLIP4Clip | 41.85 | 74.44 | 84.84 | 2.0 | 6.84 |
| +AvgPool(ratio=0.1) | 41.83 | 74.47 | 84.84 | 2.0 | 6.84 |
| +AvgPool(ratio=0.25) | 41.80 | 74.44 | 84.84 | 2.0 | 6.84 |
| +AvgPool(ratio=0.5) | 41.85 | 74.44 | 84.84 | 2.0 | 6.84 |
| +AvgPool(ratio=1) | 41.88 | 74.44 | 84.84 | 2.0 | 6.84 |
| +InvGC | 41.90 | 74.40 | 84.86 | 2.0 | 6.84 |
| +InvGC w/LocalAdj | 43.23 | 75.58 | 85.74 | 2.0 | 6.82 |
| X-CLIP | 46.25 | 76.02 | 86.05 | 2.0 | 6.37 |
| +AvgPool(ratio=0.1) | 46.47 | 75.94 | 86.05 | 2.0 | 6.38 |
| +AvgPool(ratio=0.25) | 46.38 | 75.98 | 86.01 | 2.0 | 6.38 |
| +AvgPool(ratio=0.5) | 46.47 | 75.94 | 86.01 | 2.0 | 6.38 |
| +AvgPool(ratio=1) | 46.43 | 75.89 | 86.01 | 2.0 | 6.38 |
| +InvGC | 46.43 | 75.68 | 86.22 | 2.0 | **6.35** |
| +InvGC w/LocalAdj | 47.82 | 76.46 | 86.36 | 2.0 | 6.91 |

Table 7: Retrieval performance on MSVD. Best in **Bold** and the second best is underlined.

| Normalization | Text-to-Video Retrieval | | | | |
| | R@1 | R@5 | R@10 | MdR | MnR |
|---|---|---|---|---|---|
| CLIP4Clip | 44.64 | 74.66 | 83.99 | 2.0 | 10.32 |
| +AvgPool(ratio=0.1) | 44.87 | 73.89 | 83.07 | 2.0 | 11.93 |
| +AvgPool(ratio=0.25) | 45.06 | 74.04 | 83.49 | 2.0 | 11.29 |
| +AvgPool(ratio=0.5) | 45.12 | 74.32 | 83.66 | 2.0 | 10.76 |
| +AvgPool(ratio=1) | 45.09 | 74.54 | 83.91 | 2.0 | 10.45 |
| +InvGC | 45.43 | 74.82 | 84.00 | 2.0 | 10.42 |
| +InvGC w/LocalAdj | 45.73 | 75.53 | 84.37 | 2.0 | 10.42 |
| CLIP2Video | 47.05 | 76.97 | 85.59 | 2.0 | 9.53 |
| +AvgPool(ratio=0.1) | 47.04 | 76.98 | 85.61 | 2.0 | 9.54 |
| +AvgPool(ratio=0.25) | 47.07 | 76.98 | 85.61 | 2.0 | 9.54 |
| +AvgPool(ratio=0.5) | 47.06 | 76.98 | 85.62 | 2.0 | 9.53 |
| +AvgPool(ratio=1) | 47.06 | 76.97 | 85.61 | 2.0 | 9.53 |
| +InvGC | 47.09 | 77.00 | 85.64 | 2.0 | 9.48 |
| +InvGC w/LocalAdj | 47.47 | 77.46 | 85.84 | 2.0 | 9.53 |
| X-CLIP | 46.31 | 76.84 | 85.31 | 2.0 | 9.59 |
| +AvgPool(ratio=0.1) | 46.36 | 76.70 | 85.28 | 2.0 | 9.66 |
| +AvgPool(ratio=0.25) | 46.33 | 76.71 | 85.28 | 2.0 | 9.64 |
| +AvgPool(ratio=0.5) | 46.35 | 76.71 | 85.25 | 2.0 | 9.62 |
| +AvgPool(ratio=1) | 46.41 | 76.77 | 85.27 | 2.0 | 9.60 |
| +InvGC | 46.82 | 76.69 | 85.38 | 2.0 | 9.63 |
| +InvGC w/LocalAdj | 46.49 | 76.82 | 85.29 | 2.0 | 9.63 |

Table 8: Text-audio retrieval performance on CLOTHO. Best in **Bold** and the second best is underlined. '*' refers to the results direct copied from (Koepke et al., 2022). MoEE (Miech et al., 2020), MMT (Gabeur et al., 2020) are two methods used for audio-text retrieval.

| Method | Normalization | Text-to-Audio Retrieval | | | | |
| | | R@1↑ | R@5↑ | R@10↑ | MdR↓ | MnR↓ |
|---|---|---|---|---|---|---|
| MoEE* | | 6.00 | 20.80 | 32.30 | 23.0 | 60.20 |
| MMT* | | 6.50 | 21.60 | 66.90 | 23.0 | 67.70 |
| AR-CE | | 6.27 | 22.32 | 33.30 | 23.0 | 58.96 |
| | +AvgPool(ratio=0.1) | 6.37 | 22.33 | 33.59 | 23.0 | 65.57 |
| | +AvgPool(ratio=0.25) | 6.47 | 22.12 | 33.58 | 22.0 | 59.84 |
| | +AvgPool(ratio=0.5) | 6.39 | 22.24 | 33.30 | 22.0 | 59.14 |
| | +AvgPool(ratio=1) | 6.28 | 22.32 | 33.30 | 23.0 | 58.95 |
| | +InvGC | **6.81** | 22.14 | **34.72** | **21.0** | 61.98 |
| | +InvGC w/LocalAdj | 6.58 | **22.35** | 34.64 | 22.0 | **58.51** |

**Text-Image Retrieval.** Results are presented in Tables 3 and 4. We observe that one of our methods achieves the best performance on R@1, R@5, and R@10 by a large margin. When evaluated on the CLIP method, InvGC w/LocalAdj outperforms the baselines on both the MSCOCO and Flickr30k datasets, improving R@1 and R@5 by at least 2% compared to all baselines.

**Text-Video Retrieval.** Results are presented in Tables 5 to 7. We can also conclude that one of our methods achieves the best performance on R@1,

Table 9: Text-audio retrieval performance on Audio-Caps. Best in **Bold** and the second best is underlined. '*' refers to the results direct copied from Koepke et al. (2022). MoEE (Miech et al., 2020), MMT (Gabeur et al., 2020) are two methods used for audio-text retrieval.

| Method | Normalization | Text-to-Audio Retrieval | | | | |
|---|---|---|---|---|---|---|
| | | R@1 ↑ | R@5 ↑ | R@10 ↑ | MdR ↓ | MnR ↓ |
| MoEE* | | 23.00 | 55.70 | 71.00 | 4.0 | 16.30 |
| MMT* | | 36.10 | 72.00 | 84.50 | 2.3 | 7.50 |
| AR-CE | | 22.33 | 54.49 | 70.54 | 5.0 | **15.89** |
| | +AVGPOOL(ratio=0.1) | 22.33 | 54.49 | 70.54 | 5.0 | **15.89** |
| | +AVGPOOL(ratio=0.25) | 22.33 | 54.49 | 70.54 | 5.0 | **15.89** |
| | +AVGPOOL(ratio=0.5) | 22.33 | 54.49 | 70.54 | 5.0 | **15.89** |
| | +AVGPOOL(ratio=1) | 22.33 | 54.49 | 70.54 | 5.0 | **15.89** |
| | +INVGC | 22.33 | 54.46 | **70.56** | 5.0 | **15.89** |
| | +INVGC w/LOCALADJ | **24.07** | **55.69** | 70.20 | **4.0** | 16.54 |

R@5, and R@10. Specifically, on the ActivityNet dataset, INVGC w/LOCALADJ shows excellent performance with both CLIP4CLIP and X-CLIP methods, significantly outperforming all the baselines on R@1 and R@5 roughly by 2%.

**Text-Audio Retrieval.** Results are presented in Tables 8 and 9. On the CLOTHO dataset, INVGC exhibits significantly better performance compared to all the baselines while INVGC w/LOCALADJ achieves the best results on the AudioCaps dataset.

In summary, our experiments demonstrate that employing INVGC consistently improves retrieval performance across different datasets and retrieval tasks. The models with INVGC demonstrate better accuracy and ranking in retrieving relevant videos, images, and textual descriptions based on given queries. The complete results of retrieval performance can be found in Appendix C.6.

## 5 Conclusion

This paper addressed representation degeneration problem in cross-modal retrieval, which led to a decrease in retrieval performance. The representation degeneration problem was validated across multiple benchmarks and methods. To alleviate this issue, we proposed a novel method called INVGC, inspired by graph convolution and average pooling. The method established a graph topology structure within the datasets and applied graph convolution in an inverse form with subtraction over the neighborhood. Additionally, we designed the adjacency matrix, LOCALADJ, that only leveraged the nearest neighbors of each data point rather than the entire dataset, resulting in a more effective and efficient method, INVGC w/LOCALADJ. Both INVGC and INVGC w/LOCALADJ were validated through theoretical analysis and demonstrated their ability to separate representations. Finally, extensive experi-

ments on various cross-modal benchmarks showed that both of our methods successfully alleviated the problem of representation degeneration and, as a result, improved retrieval performance.

## Limitations

First, although INVGC has been validated through theoretical analysis and demonstrated its efficacy in separating representations, its performance may vary across different datasets and modalities. The effectiveness of our method might be influenced by variations in dataset characteristics, such as data distribution, scale, and complexity. Further investigation and experimentation on a wider range of datasets are needed to fully understand the generalizability of INVGC and its performance under diverse conditions.

Second, while our method shows promising results in alleviating the representation degeneration problem, it is worth noting that cross-modal retrieval tasks can still pose challenges due to inherent differences in modalities. Variations in feature spaces, data modalities, and semantic gaps between modalities may limit the overall retrieval performance. Future research efforts should focus on exploring complementary techniques, such as multimodal fusion, attention mechanisms, or domain adaptation, to further enhance the retrieval accuracy and alleviate representation degeneration problem.

In the future, it would be interesting to explore the performance of INVGC on diverse datasets, the challenges associated with cross-modal differences, and better definitions or metrics for measuring representation degeneration problem.

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

# A  Related Work

We review prior work in cross-modal retrieval and representation degeneration, which are the two most related areas to our work.

**Cross-modal Retrieval.** The goal of cross-modal retrieval is to learn a common representation space, where the similarity between samples from different modalities can be directly measured. Recently, inspired by the success of deep learning (Devlin et al., 2019; He et al., 2016), numerous methods based on deep neural networks have been proposed for image-text retrieval (Radford et al., 2021), video-text retrieval (Luo et al., 2022), and audio-text retrieval (Oncescu et al., 2021). Further, to learn a better representation space, vision-language pretraining (Gan et al., 2020; Li et al., 2020; Singh et al., 2022) on large-scale unlabeled cross-modal data has been widely employed and have shown promising performance. Motivated by this, recent works have attempted to pretrain or fine-tune cross-modal retrieval models, e.g., image-text retrieval (Radford et al., 2021; Li et al., 2020), video-text retrieval (Chen et al., 2020; Cheng et al., 2021; Gao et al., 2021; Gorti et al., 2022; Lei et al., 2021; Ma et al., 2022; Park et al., 2022; Wang et al., 2022a,b; Zhao et al., 2022), and audio-text retrieval (Oncescu et al., 2021) in an end-to-end manner.

In contrast to the methods that focus on improving the representation learning ability, another line of cross-modal retrieval research has focused on improving the effectiveness of retrieval, including k-d trees (Bellman, 2015), re-ranking (Zhong et al., 2017; Miech et al., 2021), query expansion (Chen et al., 2020), vector compression schemes based on binary codes (Su et al., 2019; Liong et al., 2017) and quantization (Gong et al., 2013) that help address the curse of dimensionality (Keogh and Mueen, 2017). However, a recent study (Liang et al., 2022) shows that the representation degeneration problem has significantly affected the performance of multi-modal learning. To investigate the influence of representation degeneration problem in the cross-modal retrieval, we show that representation degeneration problem widely exists in different datasets and models.

**Representation degeneration**. The representation degeneration problem was first introduced in the natural language processing (NLP) area (Gao et al., 2019). It was found that when training a model for natural language generation tasks

through likelihood maximization with the weight-tying trick, especially with large training datasets, many of the learned word embeddings tend to degenerate and be distributed into a narrow cone. This limitation largely reduces the representation power of word embeddings. The representation degeneration problem leads to an increase in the overall similarity between token embeddings, which has a negative effect on the performance of the models. It was noted that Laplacian regularization can address this problem better than cosine regularization through theoretical proof and empirical experiments (Zhang et al., 2020). Subsequent work highlighted (Yu et al., 2022) that the training dynamics of the token embeddings focus on rare token embedding which leads to the degeneration problem for all tokens. To this end, they use adaptive gradient gating which gates the specific part of the gradient for rare token embeddings and thus better alleviates the data degeneration problem.

Though representation degeneration has been explored in NLP, it remains unexplored in multi-modal learning for a long time. A recent work (Liang et al., 2022) shows that the representation generated by a common deep neural network is restricted to a narrow cone and consequently, with two modality-dependent encoders, the representations from the two modalities are clearly apart during the whole training procedure. Further, they also show that varying the modality gap distance has a significant impact on improving the model's downstream zero-shot classification performance and fairness.

To step forward towards better representation learning in cross-modal retrieval, different from the previous methods in NLP which focus on addressing this problem in the training procedure, we propose a novel method, namely INVGC, which proposes to avoid representation degeneration in a post-processing manner. Inspired by the representation aggregation induced by graph convolution (Keogh and Mueen, 2017; Baranwal et al., 2023), we utilize the graph convolution in an opposite way to separate the data points that share similar representation. As the first method in solving the representation degeneration problem in cross-modal retrieval, INVGC does not require retraining the model or any other time-consuming operation. INVGC achieves better retrieval performance with a larger margin between different representations compared to the baselines and does this faster.

## B  Elaborations on Methodologies

### B.1  Graph Convolution

Graph convolution is a mathematical operation that transforms the features of nodes in a graph based on their local neighborhoods. The objective is to learn a function of signals/features on a graph, which takes into account the graph structure and the node features. It can be regarded as a generalization of convolutions to non-Euclidean data (Bruna et al., 2014). The operation is first introduced and popularized in the work of Graph Convolution Networks (Kipf and Welling, 2017), which is considered one of the most seminal papers in the area of graph learning.

The main idea behind a graph convolution operation is to generate a new representation for each node that captures the local neighborhood information around it. This is usually achieved by aggregating feature information from a node's direct neighbors, sometimes including the node itself. Formally, it can be defined as a special case of a Message Passing Network (MPNN), in which vector messages are exchanged between nodes and updated using neural networks (Gilmer et al., 2017). The basic operation of MPNN can be expressed as (Hamilton, 2020)

$$
\begin{aligned}
\mathbf{x}_i^{(k+1)} &= \mathrm{UPDATE}^{(k)}\left(\mathbf{x}_i^{(k)},\right.\\
&\quad \left.\mathrm{AGGREGATE}^{(k)}\left(\left\{\mathbf{x}_j^{(k)}, \forall j \in \mathcal{N}(i)\right\}\right)\right)\\
&= \mathrm{UPDATE}^{(k)}\left(\mathbf{x}_i^{(k)}, \mathbf{m}_{\mathcal{N}(i)}^{(k)}\right),
\end{aligned}
$$

where UPDATE and AGGREGATE are arbitrary differentiable functions (i.e., neural networks) and $\mathbf{x}_i^{(k)}$ is the embedding(representation) of node $i$ at $k$-th iteration. $\mathbf{m}_{\mathcal{N}(i)}$ is the "message" that is aggregated from $i$'s graph neighborhood $\mathcal{N}(i)$.

In this study, we adopt a simple message passing operator that performs non-trainable propagation since we want to propose a post-processing method with any training. The adopt operator is actually the backbone of multiple GNN studies, which can be expressed as (GraphConv),

$$
\mathbf{x}_i' = \bigoplus_{j \in \mathcal{N}(i)} e_{ji} \cdot \mathbf{x}_j\,,
$$

where $\bigoplus$ defines a custom aggregation scheme. $\mathbf{x}_i'$ is updated representation of node $i$ and $e_{ji}$ is edge weight between node $i$ and $j$.

For the sake of simplicity and explainability, we concretize the above operation only with simple addition and self-loop, as follows,

$$
\mathbf{x_i}' = \mathbf{x_i} + \sum_j A_{ij}\mathbf{x_j}\,. \tag{10}
$$

Note that the only difference with Equation (10) and inverse convolution Equation (3) we apply in the study is that addiction is replaced with subtraction, leading to the name 'inverse'.

### B.2  Average Pooling in CNN

Average pooling is one type of pooling layers that conducts dimensionality reduction, reducing the number of parameters in the input (Goodfellow et al., 2016). Similar to the convolutional layer, the pooling operation sweeps a filter across the entire input, but the difference is that this filter does not have any weights. Instead, the kernel applies an aggregation function to the values within the receptive field. Specifically, the adopted average pooling calculates the average value within the receptive field of the filter as it moves across the input. Here in this study, the receptive field is subject to the size of the neighborhood of each data point.

### B.3  Distribution of Data Representation

While we keep using the discrete gallery and query set like in Equation (3), they can be regarded as sampled results from the hidden continuous distribution dependent on the intrinsic properties of the corresponding dataset and the representation learning method applied. Therefore, ideally, two gallery sets sampled from the same dataset will probably have different $\Delta_{deg}$ scores even with the same representation learning method due to the variance introduced by the sampling process. That is to say, performing the same inverse convolution operation on the two sampled gallery sets might have quite different effects, especially when the sample size is small. Also, since the proposed method in this study is a post-processing approach without any training, we need to control the magnitude of the convolution with the help of some hyperparameters. The reason for doing this is to control the change in the representation of the gallery data that has already been aligned with the embedding space of query data by the representation learning model. Given the sampled gallery set is small, this means a very large variance in the value of the best hyperparameters as well.

Unfortunately, it is quite common in practice that we only have a small sampled gallery set. Usually, when cross-modal retrieval is carried out, we constantly cut down the size of the gallery set with the help of some pre-ranking or indexing techniques. This process can be somehow regarded as sampling a set from the distribution that is empirically represented by the whole gallery set. Also, during the evaluation of any method on various datasets, the size of the test or evaluation gallery set is typically much smaller compared to the training set. Both cases make the result of the proposed methods subjected to potentially large variance.

However, it would be more promising that our method is generally stable and robust to any size of the gallery set. The ideal case would be that we can perform the inverse convolution similar to Equation (1) but based on the continuous distribution, where $P(\mathbf{x_j})$ is data point $\mathbf{x_j}$ to be sampled from the hidden distribution, as follows,

$$\mathbf{x_i}' = \mathbf{x_i} - r \int_{\mathbf{x_j}} S_{ij} \, P(\mathbf{x_j}) \mathbf{x_j} \,. \qquad (11)$$

However, it is impossible to have exact access to this hidden distribution in practice. The best approximation is the training (or validation) gallery set $\hat{G}$ since it is the largest one we can obtain. Therefore, we can perform the inverse convolution on $\hat{G}$. Note that the distribution of the query set $\hat{Q}$ should theoretically be similar to that of $\hat{G}$ as this a basic assumption in machine learning (Bogolin et al., 2022). Therefore, it is possible to combine the (train or validation) gallery set $\hat{G}$ and the (train or validation) query set $\hat{Q}$ to be the even better estimation of the hidden distribution. Thus, we go on to refine INVGC as in Equation (4),

In general, with reasonable and general assumptions, we strike the importance of utilizing the data of both the modality from the training set when we want to capture a more accurate and stable distribution of data representation. The idea is not bound to the proposed methods and can be adopted by any future work on the post-processing of cross-modal retrieval tasks.

## C  Experiments

### C.1  Datasets Details

The experiments are conducted on eight cross-modal benchmarks, which include four video-text retrieval benchmarks (MSR-VTT (Xu et al.,

2016), MSVD (Chen and Dolan, 2011), ActivityNet (Fabian Caba Heilbron and Niebles, 2015), and DiDemo (Hendricks et al., 2017)), two image-text retrieval benchmarks (MSCOCO (Lin et al., 2014) and Flickr30k (Plummer et al., 2017)), as well as two audio-text retrieval benchmarks (AudioCaps (Kim et al., 2019) and CLOTHO (Drossos et al., 2020)). The details of the datasets are presented below:

- **MSR-VTT** (Xu et al., 2016): Comprises approximately 10k videos, each accompanied by 20 captions. For text-video retrieval, we follow the protocol set by previous works (Liu et al., 2019; Croitoru et al., 2021; Luo et al., 2022; Ma et al., 2022; Park et al., 2022), using both the official (full) split and the 1k-A split. The full split includes 2,990 videos for testing and 497 for validation, whereas the 1k-A split has 1,000 videos for testing and around 9,000 for training.

- **MSVD** (Chen and Dolan, 2011): Contains 1,970 videos and about 80k captions. The standard split used in prior works (Liu et al., 2019; Croitoru et al., 2021; Luo et al., 2022; Park et al., 2022) is adopted for reporting results, which includes 1,200 videos for training, 100 for validation, and 670 for testing.

- **ActivityNet** (Fabian Caba Heilbron and Niebles, 2015): Contains 20k videos and approximately 100K descriptive sentences. These videos are extracted from YouTube. We employ a paragraph video retrieval setup as defined in prior works (Liu et al., 2019; Croitoru et al., 2021; Luo et al., 2022; Park et al., 2022). We report results on the val1 split. The training split includes 10,009 videos, with 4,917 videos allocated for testing.

- **DiDemo** (Hendricks et al., 2017): Includes over 10,000 personal videos, each lasting between 25-30 seconds, along with over 40,000 localized text descriptions. The videos are divided into training (8,395), validation (1,065), and testing (1,004) sets.

- **MSCOCO** (Lin et al., 2014): Consists of 123k images, each accompanied by 5 captions. The 5k split is used for evaluation.

- **Flickr30k** (Plummer et al., 2017): This dataset contains 31,000 images collected from

Table 10: Implmentation details.

| Method | Public Code | Public Weights |
|---|---|---|
| CE+ | Link | MSR-VTT, MSVD, DiDeMo, and ActivityNet |
| TT-CE+ | Link | MSR-VTT, MSVD, DiDeMo, and ActivityNet |
| CLIP4Clip | Link | N/A |
| CLIP2Video | Link | MSR-VTT and MSVD |
| X-CLIP | Link | N/A |
| CLIP | Link | CLIP |
| Oscar | Link | MSCOCO and Flickr30k |
| AR-CE | Link | AudioCaps and CLOTHO |

Flickr, each accompanied by 5 reference sentences provided by human annotators.

- **AudioCaps** (Kim et al., 2019): Comprises sound samples with event descriptions. We adopt the same setup as prior work (Koepke et al., 2022) where 49,291 samples are used for training, 428 for validation, and 816 for testing.

- **CLOTHO** (Drossos et al., 2020): Comprises of 4,981 audio samples of 15 to 30 seconds in duration and 24,905 captions of eight to 20 words in length (five captions for each audio sample).

## C.2 Experiment Details

The public codes and weights of all the tasks in this study are summarized in Table 10. Note that there is no available trained model for CLIP and X-CLIP. Therefore, we train both models on a single A100 GPU with the hyperparameters recommended by the original studies.

## C.3 Prevalence of Representation Degeneration Problem across Datasets and Methods

To show that the representation degeneration problem prevails in all datasets, methods, and tasks, we perform the same analysis in Figure 1a. We uniformly sample a subset of the gallery set of both retrieval tasks (i.e., text to other modality or other modality to text, other modalities can be video, image, or audio depending on the dataset), and perform PCA upon it to reduce the dimension of the representations down to 2, which is the first two principal dimensions. The results are presented in Figure 5. Note that $\Delta_{deg}$ (i.e., the degree of representation degeneration problem) included in each figure is the one for the complete gallery set instead of the sampled set used in the figure.

For this qualitative but very intuitive study, We firmly validate again that almost all the data representations gathered in a very narrow cone in the embedding space for basically all the datasets, methods, and tasks. Also, though subject to the difference between datasets and methods, we can witness that a more convex-shaped distribution usually generally leads to a larger $\Delta_{deg}$.

The results imply the universality of the representation degeneration problem. More quantitative results can be found in **RQ1** in Section 4.3 and in the **Continuation on RQ1** section (Appendix C.7).

## C.4 Detail Setting of AVGPOOL

Following Equation (9), the adjacency matrices $\mathcal{S}^g_{\text{pool}}$ and $\mathcal{S}^q_{\text{pool}}$ of AVGPOOL can be presented as,

$$\mathcal{S}^g_{\text{pool}}(i,j) = \begin{cases} 1, \text{sim}(\mathbf{g_i}, \hat{\mathbf{g}}_\mathbf{j}) \geq P_i(\hat{G}, p) \\ 0, \text{else} \end{cases}$$

and also,

$$\mathcal{S}^q_{\text{pool}}(i,j) = \begin{cases} 1, \text{sim}(\mathbf{g_i}, \hat{\mathbf{q}}_\mathbf{j}) \geq P_i(\hat{Q}, p) \\ 0, \text{else}, \end{cases}$$

where $P_i(\hat{G}, p)$ and $P_i(\hat{Q}, p)$ are the value of $p$-percentage largest similarity between node $i$ and all the nodes in set $\hat{G}$ and $\hat{Q}$, respectively. We include four values of $p$, namely $[10, 25, 50, 100]$, to obtain a solid benchmark. All four benchmarks for the same task share identical hyperparameters $r_g$ and $r_q$. Note that the worst scenario of tuning for AVGPOOL is when there is no hyperparameter that can enable AVGPOOL to beat the baseline performance of the original retrieval model. But it doesn't happen during the experiment since we can always locate a pair of hyperparameters with a little effort with which AVGPOOL with at least one of four selected $p$ values can outperform the baseline on at least one of the evaluation metrics mentioned in Section 4.2. This indicates the effectiveness of the idea of inverse convolution.

## C.5 Computational Complexity

In the section, we discuss the computation complexity of the proposed INVGC and INVGC w/LOCALADJ. As noted Section 2.1, $N_g$ is the size of gallery data, and $N_{\hat{Q}}$ and $N_{\hat{G}}$ are the size of the training (or validation) query and gallery set, respectively. We also assume $N_q$ is the size of the query.

Both methods need to precompute two adjacency matrices between the gallery set and both sets of training (or validation) data before performing the convolution, which costs $\mathcal{O}(N_g(N_{\hat{G}} + N_{\hat{Q}}))$ as

Table 11: Retrieval performance on MSCOCO (5k split). Best in **Bold** and the second best is underlined.

| Normalization | Text-to-Image Retrieval | | | | | Image-to-Text Retrieval | | | | |
|---|---|---|---|---|---|---|---|---|---|---|
| | R@1↑ | R@5↑ | R@10↑ | MdR↓ | MnR↓ | R@1↑ | R@5↑ | R@10↑ | MdR↓ | MnR↓ |
| CLIP | 30.34 | 54.74 | 66.08 | 4.0 | 25.39 | 50.04 | 74.80 | 83.38 | **1.0** | 9.22 |
| +AvgPool(ratio=0.1) | 30.37 | 54.77 | 66.14 | 4.0 | 25.36 | 49.98 | 75.08 | 83.34 | 2.0 | 9.20 |
| +AvgPool(ratio=0.25) | 30.37 | 54.77 | 66.14 | 4.0 | 25.36 | 49.98 | 75.06 | 83.30 | 2.0 | 9.20 |
| +AvgPool(ratio=0.5) | 30.38 | 54.77 | 66.10 | 4.0 | 25.38 | 49.98 | 75.10 | 83.34 | 2.0 | 9.20 |
| +AvgPool(ratio=1) | 30.39 | 54.77 | 66.11 | 4.0 | 25.38 | 49.98 | 75.06 | 83.36 | 2.0 | 9.20 |
| +InvGC | 32.70 | 57.53 | **68.24** | 4.0 | **24.35** | 51.04 | 75.18 | 83.24 | **1.0** | 8.93 |
| +InvGC w/LocalAdj | **33.11** | 57.49 | 68.19 | 4.0 | 28.95 | **52.26** | **76.42** | **84.32** | **1.0** | 8.83 |
| Oscar | 52.50 | 80.03 | **87.96** | 1.0 | 10.68 | 66.74 | 89.98 | 94.98 | 1.0 | 2.95 |
| +AvgPool(ratio=0.1) | 52.52 | 80.04 | 87.95 | 1.0 | 10.70 | 66.98 | 89.98 | 95.00 | 1.0 | 2.96 |
| +AvgPool(ratio=0.25) | 52.52 | 80.03 | **87.96** | 1.0 | 10.68 | 66.98 | 89.96 | 94.96 | 1.0 | 2.95 |
| +AvgPool(ratio=0.5) | 52.51 | 80.00 | **87.96** | 1.0 | **10.67** | 66.94 | 89.92 | 94.94 | 1.0 | 2.95 |
| +AvgPool(ratio=1) | 52.50 | 80.02 | **87.96** | 1.0 | 10.68 | 66.70 | 89.90 | 95.00 | 1.0 | 2.96 |
| +InvGC | 52.63 | **80.05** | **87.96** | 1.0 | 10.72 | **67.90** | 89.96 | **95.22** | 1.0 | **2.92** |
| +InvGC w/LocalAdj | **52.93** | **80.05** | 87.78 | 1.0 | 11.09 | 67.68 | **90.24** | 95.20 | 1.0 | 2.94 |

Table 12: Retrieval performance on Flickr30k. Best in **Bold** and the second best is underlined.

| Normalization | Text-to-Image Retrieval | | | | | Image-to-Text Retrieval | | | | |
|---|---|---|---|---|---|---|---|---|---|---|
| | R@1↑ | R@5↑ | R@10↑ | MdR↓ | MnR↓ | R@1↑ | R@5↑ | R@10↑ | MdR↓ | MnR↓ |
| CLIP | 58.98 | 83.48 | 90.14 | 1.0 | 6.04 | 78.10 | 94.90 | 98.10 | 1.0 | 1.98 |
| +AvgPool(ratio=0.1) | 59.10 | **83.56** | 90.18 | 1.0 | 6.04 | 78.30 | 95.00 | 98.20 | 1.0 | **1.97** |
| +AvgPool(ratio=0.25) | 59.10 | **83.56** | 90.18 | 1.0 | 6.04 | 78.40 | 95.00 | 98.20 | 1.0 | **1.97** |
| +AvgPool(ratio=0.5) | 59.10 | 83.54 | 90.18 | 1.0 | 6.05 | 78.40 | 95.00 | 98.20 | 1.0 | **1.97** |
| +AvgPool(ratio=1) | 59.10 | 83.54 | 90.18 | 1.0 | 6.04 | 78.40 | 95.00 | 98.20 | 1.0 | 1.98 |
| +InvGC | 60.18 | 85.30 | **91.20** | 1.0 | **5.52** | 78.50 | 95.10 | 98.20 | 1.0 | 1.98 |
| +InvGC w/LocalAdj | **60.48** | 85.30 | 91.10 | 1.0 | 5.59 | **80.20** | 95.10 | **98.40** | 1.0 | **1.97** |
| Oscar | 71.60 | 91.50 | 94.96 | 1.0 | **4.24** | 86.30 | 96.80 | 98.60 | 1.0 | **1.58** |
| +AvgPool(ratio=0.1) | 71.62 | 91.44 | 94.92 | 1.0 | 4.25 | 86.50 | 96.70 | 98.50 | 1.0 | 1.63 |
| +AvgPool(ratio=0.25) | 71.66 | 91.50 | 94.94 | 1.0 | **4.24** | 86.00 | 96.90 | 98.60 | 1.0 | 1.62 |
| +AvgPool(ratio=0.5) | 71.66 | 91.50 | 94.92 | 1.0 | **4.24** | 86.00 | 97.00 | 98.60 | 1.0 | 1.60 |
| +AvgPool(ratio=1) | 71.62 | 91.52 | 94.96 | 1.0 | **4.24** | 86.10 | 97.00 | 98.60 | 1.0 | **1.58** |
| +InvGC | 71.68 | 91.46 | **95.06** | 1.0 | 4.25 | **86.80** | 96.80 | 98.40 | 1.0 | 1.61 |
| +InvGC w/LocalAdj | **71.74** | **91.56** | 94.98 | 1.0 | 4.29 | 86.60 | **97.10** | **98.70** | 1.0 | **1.58** |

matrix multiplication. Then, the matrix multiplication of inverse convolution operation also incurs a cost of $\mathcal{O}(N_g(N_{\hat{G}} + N_{\hat{Q}}))$ for INVGC and $\mathcal{O}(k^{-1}N_g(N_{\hat{G}} + N_{\hat{Q}}))$ for INVGC w/LOCALADJ, for the pruned matrix LOCALADJ. Note that if we choose a small $k$ for LOCALADJ like $1\%$ in the empirical study, the convolution step can be much faster in practice (though the complexity doesn't change due to the pre-computation step).

After performing the proposed methods, we need another $\mathcal{O}(N_q N_g)$ time to calculate the similarity between the query and gallery set when performing the inference, given that we don't consider any advanced trick to perform argmax.

In sum, both INVGC and INVGC w/LOCALADJ incur computational cost of $\mathcal{O}(N_g(N_{\hat{G}} + N_{\hat{Q}}))$ for inverse graph convolution (before inference) and $\mathcal{O}(N_q N_g)$ for inference.

## C.6 More Quantitative Results

The results on retrieval performance for all the datasets with various methods are presented in Tables 11 to 18. The results indicate that both proposed methods achieve significantly better retrieval performance on all the tasks compared to the original baseline and AVGPOOL benchmarks.

**Text-Image and Image-Text Retrieval.** Results are presented in Tables 11 and 12.

On the MSCOCO (5k split) dataset, the INVGC w/LOCALADJ method outperforms other methods in both text-to-image retrieval and image-to-text Retrieval for both CLIP and Oscar models. Specifically, for the CLIP model, INVGC w/LOCALADJ achieves the best R@1 and R@5 in text-to-image retrieval and the best R@1, R@5, R@10, and MnR scores in image-to-text retrieval. Similarly, for the Oscar model, INVGC w/LOCALADJ also delivers the best results, with the best R@1 and R@5 in

Table 13: Retrieval performance on MSR-VTT (full split and 1k split). Best in **Bold** and the second best is underlined.

| Normalization | | Text-to-Video Retrieval | | | | | Video-to-Text Retrieval | | | | |
|---|---|---|---|---|---|---|---|---|---|---|---|
| | | R@1 | R@5 | R@10 | MdR | MnR | R@1 | R@5 | R@10 | MdR | MnR |
| *MSR-VTT (full split)* | | | | | | | | | | | |
| CE+ | | 13.51 | 36.01 | 48.75 | 11.0 | **70.28** | 21.61 | 50.57 | 63.48 | 5.0 | 22.62 |
| | +AVGPOOL(ratio=0.1) | 13.52 | 36.02 | 48.78 | 11.0 | 70.36 | 21.57 | 50.54 | 63.44 | 5.0 | 22.63 |
| | +AVGPOOL(ratio=0.25) | 13.50 | 36.00 | 48.76 | 11.0 | 70.33 | 21.57 | 50.57 | 63.48 | 5.0 | 22.62 |
| | +AVGPOOL(ratio=0.5) | 13.51 | 35.99 | 48.76 | 11.0 | 70.31 | 21.57 | 50.57 | 63.48 | 5.0 | 22.62 |
| | +AVGPOOL(ratio=1) | 13.51 | 36.01 | 48.75 | 11.0 | **70.28** | 21.61 | 50.57 | 63.48 | 5.0 | 22.62 |
| | +INVGC | 13.57 | 36.01 | 48.74 | 11.0 | 71.34 | **22.47** | 50.67 | 63.68 | 5.0 | **22.42** |
| | +INVGC w/LOCALADJ | **13.83** | **36.54** | **49.18** | 11.0 | 71.25 | 22.24 | **51.37** | **64.75** | 5.0 | 23.86 |
| TT-CE+ | | 14.51 | 37.58 | 50.39 | 10.0 | 64.29 | 24.11 | 53.71 | 67.39 | 5.0 | 20.20 |
| | +AVGPOOL(ratio=0.1) | 14.51 | 37.59 | 50.39 | 10.0 | 64.29 | 24.21 | 53.48 | 67.39 | 5.0 | 20.22 |
| | +AVGPOOL(ratio=0.25) | 14.51 | 37.58 | 50.39 | 10.0 | 64.29 | 24.18 | 53.58 | 67.36 | 5.0 | 20.23 |
| | +AVGPOOL(ratio=0.5) | 14.51 | 37.58 | 50.39 | 10.0 | 64.29 | 24.18 | 53.55 | 67.53 | 5.0 | 20.22 |
| | +AVGPOOL(ratio=1) | 14.51 | 37.58 | 50.39 | 10.0 | 64.29 | 24.11 | 53.71 | 67.39 | 5.0 | 20.20 |
| | +INVGC | 14.62 | 37.61 | 50.50 | 10.0 | 64.89 | 25.15 | 53.78 | 67.63 | 5.0 | 21.43 |
| | +INVGC w/LOCALADJ | **15.08** | **38.48** | **51.57** | 10.0 | 64.19 | **26.05** | **54.72** | **68.73** | 4.5 | **19.83** |
| *MSR-VTT (1k split)* | | | | | | | | | | | |
| CLIP4Clip | | 44.10 | 71.70 | 81.40 | 2.0 | 15.51 | 42.09 | 71.25 | 81.23 | 2.0 | 12.02 |
| | +AVGPOOL(ratio=0.1) | 44.20 | 71.60 | 81.50 | 2.0 | 15.55 | 42.39 | 70.65 | 80.24 | 2.0 | 12.34 |
| | +AVGPOOL(ratio=0.25) | 44.20 | 71.60 | 81.50 | 2.0 | 15.55 | 42.39 | 70.55 | 79.94 | 2.0 | 12.37 |
| | +AVGPOOL(ratio=0.5) | 44.20 | 71.50 | 81.50 | 2.0 | 15.54 | 42.39 | 70.75 | 79.84 | 2.0 | 12.33 |
| | +AVGPOOL(ratio=1) | 44.10 | 71.60 | 81.50 | 2.0 | 15.52 | 42.29 | 70.75 | 80.04 | 2.0 | 12.30 |
| | +INVGC | 44.40 | **71.90** | **81.60** | 2.0 | **15.36** | **44.66** | **72.13** | **81.72** | 2.0 | **11.59** |
| | +INVGC w/LOCALADJ | **44.40** | 71.70 | 81.20 | 2.0 | 15.65 | 42.79 | 71.44 | 80.83 | 2.0 | 12.08 |
| CLIP2Video | | 46.00 | 71.60 | 81.60 | 2.0 | 14.51 | 43.87 | 72.73 | 82.51 | 2.0 | 10.20 |
| | +AVGPOOL(ratio=0.1) | 45.90 | 71.70 | 81.50 | 2.0 | 14.52 | 43.87 | 72.83 | 82.41 | 2.0 | 10.21 |
| | +AVGPOOL(ratio=0.25) | 46.10 | 71.80 | 81.50 | 2.0 | 14.53 | 43.77 | 72.73 | 82.31 | 2.0 | 10.21 |
| | +AVGPOOL(ratio=0.5) | 46.00 | 71.70 | 81.50 | 2.0 | 14.53 | 43.77 | 72.73 | 82.31 | 2.0 | 10.21 |
| | +AVGPOOL(ratio=1) | 45.90 | 71.70 | 81.50 | 2.0 | 14.52 | 43.77 | 72.73 | 82.41 | 2.0 | 10.21 |
| | +INVGC | 46.20 | 71.70 | 81.30 | 2.0 | **14.44** | 44.66 | **73.22** | **83.10** | 2.0 | **10.07** |
| | +INVGC w/LOCALADJ | **46.60** | **72.10** | **81.70** | 2.0 | 14.50 | **45.06** | 70.65 | 79.64 | 2.0 | 11.65 |
| X-CLIP | | 46.30 | 74.00 | 83.40 | 2.0 | **12.80** | 44.76 | 73.62 | 82.31 | 2.0 | 11.02 |
| | +AVGPOOL(ratio=0.1) | 46.50 | 74.00 | 83.40 | 2.0 | 12.88 | 44.92 | 73.64 | 82.23 | 2.0 | 11.07 |
| | +AVGPOOL(ratio=0.25) | 46.40 | 74.00 | **83.50** | 2.0 | 12.85 | 44.96 | 73.52 | 82.41 | 2.0 | 10.87 |
| | +AVGPOOL(ratio=0.5) | 46.30 | 74.00 | 83.30 | 2.0 | 12.83 | 44.76 | 73.42 | 82.11 | 2.0 | 10.78 |
| | +AVGPOOL(ratio=1) | 46.20 | 74.00 | 83.40 | 2.0 | 12.83 | 44.66 | 73.52 | 82.02 | 2.0 | **10.76** |
| | +INVGC | **47.30** | 74.00 | 83.30 | 2.0 | 13.42 | 46.05 | **75.40** | **83.20** | 2.0 | 10.86 |
| | +INVGC w/LOCALADJ | 47.10 | **74.20** | **83.50** | 2.0 | 13.09 | **46.15** | 71.94 | 80.83 | 2.0 | 11.95 |

text-to-image retrieval.

In the Flickr30k dataset, the INVGC w/LOCALADJ method generally shows the highest performance in both text-to-image retrieval and image-to-text retrieval across both CLIP and Oscar models.

**Text-Video and Video-Text Retrieval.** Results are presented in Tables 13 to 16.

From the table for MSR-VTT (full split), we can observe that INVGC w/LOCALADJ demonstrated the best performance in both text-to-video and video-to-text retrieval on R@1, R@5, and R@10 with both methods. The results for MSR-VTT (1k split) show that INVGC w/LOCALADJ generally achieves the best performance in text-to-video retrieval while INVGC presents superior results in video-to-text retrieval.

For the ActivityNet dataset, INVGC w/LOCALADJexhibits superior performance in text-to-video and video-to-text retrieval on R@1, R@5, and R@10 with all four methods.

For the MSVD dataset, the best-performing method for text-to-video retrieval is INVGC w/LOCALADJ since it has the best performance on R@1, R@5, and R@10 with all methods except X-CLIP. Meanwhile, INVGC method has the best

Table 14: Retrieval performance on ActivityNet. Best in **Bold** and the second best is underlined.

| | Normalization | Text-to-Video Retrieval | | | | | Video-to-Text Retrieval | | | | |
| --- | --- | --- | --- | --- | --- | --- | --- | --- | --- | --- | --- |
| | | R@1 | R@5 | R@10 | MdR | MnR | R@1 | R@5 | R@10 | MdR | MnR |
| CE+ | | 19.16 | 49.79 | 65.79 | 6.0 | 21.99 | 18.51 | 47.85 | 63.94 | 6.0 | 23.06 |
| | +AvgPool(ratio=0.1) | 19.22 | 49.77 | 65.79 | 6.0 | 21.98 | 18.63 | 47.87 | 63.92 | 6.0 | 23.03 |
| | +AvgPool(ratio=0.25) | 19.20 | 49.79 | 65.77 | 6.0 | 21.98 | 18.59 | 47.94 | 63.92 | 6.0 | 23.01 |
| | +AvgPool(ratio=0.5) | 19.20 | 49.79 | 65.77 | 6.0 | 21.98 | 18.59 | 47.90 | 63.94 | 6.0 | 23.01 |
| | +AvgPool(ratio=1) | 19.16 | 49.79 | 65.79 | 6.0 | 21.99 | 18.51 | 47.85 | 63.94 | 6.0 | 23.06 |
| | +InvGC | 19.34 | 49.58 | **66.02** | 6.0 | **21.33** | 18.71 | 48.12 | **64.10** | 6.0 | **22.75** |
| | +InvGC w/LocalAdj | **19.89** | 50.21 | 65.47 | 5.0 | 23.05 | **19.22** | 48.14 | 63.68 | 6.0 | 25.96 |
| TT-CE+ | | 23.29 | 56.42 | 73.78 | 4.0 | 13.59 | 22.49 | 56.38 | 72.67 | 4.0 | 13.90 |
| | +AvgPool(ratio=0.1) | 23.29 | 56.46 | 73.81 | 4.0 | 13.59 | 22.45 | 56.21 | 72.56 | 4.0 | 13.89 |
| | +AvgPool(ratio=0.25) | 23.27 | 56.44 | 73.81 | 4.0 | 13.59 | 22.49 | 56.25 | 72.63 | 4.0 | 13.89 |
| | +AvgPool(ratio=0.5) | 23.25 | 56.42 | 73.81 | 4.0 | 13.59 | 22.49 | 56.27 | 72.63 | 4.0 | 13.89 |
| | +AvgPool(ratio=1) | 23.29 | 56.42 | 73.78 | 4.0 | 13.59 | 22.49 | 56.38 | 72.67 | 4.0 | 13.90 |
| | +InvGC | 23.47 | 56.66 | 73.93 | 4.0 | **13.51** | 22.57 | 56.34 | 72.54 | 4.0 | **13.87** |
| | +InvGC w/LocalAdj | **23.90** | **58.12** | **74.31** | 4.0 | 13.53 | **24.51** | **57.41** | **73.34** | 4.0 | 14.59 |
| CLIP4Clip | | 41.85 | 74.44 | 84.84 | 2.0 | 6.84 | 41.62 | 74.11 | 86.12 | 2.0 | 6.81 |
| | +AvgPool(ratio=0.1) | 41.83 | 74.47 | 84.84 | 2.0 | 6.84 | 41.59 | 74.06 | 86.10 | 2.0 | 6.81 |
| | +AvgPool(ratio=0.25) | 41.80 | 74.44 | 84.84 | 2.0 | 6.84 | 41.54 | 74.09 | 86.10 | 2.0 | 6.81 |
| | +AvgPool(ratio=0.5) | 41.85 | 74.44 | 84.84 | 2.0 | 6.84 | 41.54 | 74.11 | 86.10 | 2.0 | 6.81 |
| | +AvgPool(ratio=1) | 41.88 | 74.44 | 84.84 | 2.0 | 6.84 | 41.57 | 74.11 | 86.12 | 2.0 | 6.81 |
| | +InvGC | 41.90 | 74.40 | 84.86 | 2.0 | 6.84 | 41.95 | 74.02 | 86.12 | 2.0 | 6.77 |
| | +InvGC w/LocalAdj | **43.23** | **75.58** | **85.74** | 2.0 | **6.82** | **43.20** | **75.77** | **86.67** | 2.0 | **6.63** |
| X-CLIP | | 46.25 | **76.02** | 86.05 | 2.0 | 6.37 | 45.20 | 76.07 | 86.57 | 2.0 | 6.40 |
| | +AvgPool(ratio=0.1) | 46.47 | 75.94 | 86.05 | 2.0 | 6.38 | 45.20 | 76.20 | 86.53 | 2.0 | 6.39 |
| | +AvgPool(ratio=0.25) | 46.38 | 75.98 | 86.01 | 2.0 | 6.38 | 45.29 | 76.24 | 86.53 | 2.0 | 6.38 |
| | +AvgPool(ratio=0.5) | 46.47 | 75.94 | 86.01 | 2.0 | 6.38 | 45.25 | 76.24 | 86.53 | 2.0 | 6.39 |
| | +AvgPool(ratio=1) | 46.43 | 75.89 | 86.01 | 2.0 | 6.38 | 45.15 | 76.16 | 86.53 | 2.0 | 6.40 |
| | +InvGC | 46.43 | 75.68 | 86.22 | 2.0 | **6.35** | 45.95 | 76.24 | 87.53 | 2.0 | 6.10 |
| | +InvGC w/LocalAdj | **47.82** | 76.46 | **86.36** | 2.0 | 6.91 | **48.69** | **77.90** | **87.93** | 2.0 | **6.00** |

results for video-to-text retrieval with CLIP4CLIP, CLIP2Video, and X-CLIP.

On the DiDeMo dataset, INvGC w/LocalAdj with both methods achieve the best performance in the text-to-video and video-to-text retrieval on R@1 and R@5.

**Text-Audio and Audio-Text Retrieval.** Results are presented in Tables 17 and 18. On the CLOTHO dataset, the AR-CE method enhanced with the INvGC and INvGC w/LocalAdj techniques outperforms the earlier techniques, MoEE and MMT, and all other benchmarks. Specifically, either INvGC or INvGC w/LocalAdj achieves the best results in R@1, R@5, and R@10 in both text-to-audio and audio-to-text retrieval tasks.

In the case of the AudioCaps dataset, INvGC w/LocalAdj method performs best across multiple metrics in both text-to-audio and audio-to-text retrieval tasks.

## C.7 More Ablation Study

**Continuation on RQ1: Is the data degeneration problem alleviated?**

To strengthen the conclusion that both INvGC and INvGC w/LocalAdjhave strong capability to alleviate representation degeneration problem, we conduct a comprehensive experiment on all the datasets and methods as an extension to Table 1 and Table 2. We have Tables 19 to 26 that presents the mean similarity within the test gallery set of both tasks for three scenarios, the overall mean (MeanSim), the mean between the nearest neighbor(MeanSim@1), and the mean between nearest 10 neighbors (MeanSim@10). And we have Tables 27 to 34 include the similarity score from the test gallery set to the test query set with the same evaluation metrics.

It is quite obvious that both INvGC and INvGC w/LocalAdj help decrease the similarity score on all metrics, especially on MeanSim@1(i.e., $\Delta_{deg}(G)$). Also, both methods have better per-

Table 15: Retrieval performance on MSVD. Best in **Bold** and the second best is underlined.

| | Normalization | Text-to-Video Retrieval | | | | | Video-to-Text Retrieval | | | | |
|---|---|---|---|---|---|---|---|---|---|---|---|
| | | R@1 | R@5 | R@10 | MdR | MnR | R@1 | R@5 | R@10 | MdR | MnR |
| CE+ | | 23.94 | 54.98 | 69.00 | 4.0 | 18.46 | 22.84 | 49.85 | 61.49 | 6.0 | 33.96 |
| | +AvgPool(ratio=0.1) | 24.11 | 55.26 | 69.25 | 4.0 | 18.62 | 23.88 | 50.15 | 60.15 | 5.5 | 35.54 |
| | +AvgPool(ratio=0.25) | 24.01 | 55.10 | 69.09 | 4.0 | 18.60 | 23.13 | 50.15 | 60.60 | 5.5 | 34.86 |
| | +AvgPool(ratio=0.5) | 23.95 | 55.02 | 69.05 | 4.0 | 18.52 | 23.58 | 49.85 | 60.45 | 6.0 | 34.51 |
| | +AvgPool(ratio=1) | 23.94 | 54.98 | 69.00 | 4.0 | 18.46 | 22.84 | 49.85 | 61.49 | 6.0 | 33.96 |
| | +InvGC | 24.47 | 55.43 | 69.54 | 4.0 | 25.31 | **28.51** | 51.34 | 61.49 | 5.0 | 70.09 |
| | +InvGC w/LocalAdj | **25.49** | **56.87** | **71.08** | 4.0 | **18.34** | 27.91 | 57.46 | 67.76 | 4.0 | 25.97 |
| TT-CE+ | | 24.42 | 56.20 | 70.44 | 4.0 | 17.16 | 25.22 | 55.07 | 64.63 | 4.0 | 29.94 |
| | +AvgPool(ratio=0.1) | 24.60 | 56.47 | 70.72 | 4.0 | 17.22 | 25.07 | 54.93 | 64.63 | 4.0 | 29.99 |
| | +AvgPool(ratio=0.25) | 24.56 | 56.31 | 70.51 | 4.0 | 17.21 | 25.07 | 55.37 | 64.78 | 4.0 | 29.98 |
| | +AvgPool(ratio=0.5) | 24.49 | 56.18 | 70.50 | 4.0 | 17.19 | 25.07 | 55.22 | 64.48 | 4.0 | 29.95 |
| | +AvgPool(ratio=1) | 24.42 | 56.20 | 70.44 | 4.0 | 17.16 | 25.22 | 55.07 | 64.63 | 4.0 | 29.94 |
| | +InvGC | 25.36 | 56.51 | 70.97 | 4.0 | 17.22 | **28.81** | 55.52 | 65.22 | 4.0 | 29.92 |
| | +InvGC w/LocalAdj | **26.36** | **58.34** | **72.29** | 4.0 | **16.99** | 28.06 | 57.16 | 68.36 | 4.0 | **24.49** |
| CLIP4Clip | | 44.64 | 74.66 | 83.99 | 2.0 | 10.32 | 63.13 | 79.40 | 85.37 | 1.0 | 11.02 |
| | +AvgPool(ratio=0.1) | 44.87 | 73.89 | 83.07 | 2.0 | 11.93 | 63.28 | 79.55 | 85.22 | 1.0 | 11.06 |
| | +AvgPool(ratio=0.25) | 45.06 | 74.04 | 83.49 | 2.0 | **11.29** | 62.84 | 79.55 | 85.22 | 1.0 | 10.99 |
| | +AvgPool(ratio=0.5) | 45.12 | 74.32 | 83.66 | 2.0 | 10.76 | 63.13 | 79.55 | 85.22 | 1.0 | 11.02 |
| | +AvgPool(ratio=1) | 45.09 | 74.54 | 83.91 | 2.0 | 10.45 | 62.84 | 79.55 | 85.37 | 1.0 | 11.02 |
| | +InvGC | 45.43 | 74.82 | 84.00 | 2.0 | 10.42 | **68.36** | **84.93** | **90.30** | 1.0 | **8.75** |
| | +InvGC w/LocalAdj | **45.73** | **75.53** | **84.37** | 2.0 | 10.42 | 64.33 | 80.15 | 85.67 | 1.0 | 10.77 |
| CLIP2Video | | 47.05 | 76.97 | 85.59 | 2.0 | 9.53 | 62.09 | 83.13 | 89.40 | 1.0 | 7.73 |
| | +AvgPool(ratio=0.1) | 47.04 | 76.98 | 85.61 | 2.0 | 9.54 | 62.39 | 83.73 | 89.40 | 1.0 | 7.76 |
| | +AvgPool(ratio=0.25) | 47.07 | 76.98 | 85.61 | 2.0 | 9.54 | 62.09 | 83.28 | 89.40 | 1.0 | 7.74 |
| | +AvgPool(ratio=0.5) | 47.06 | 76.98 | 85.62 | 2.0 | 9.53 | 61.94 | 83.73 | 89.25 | 1.0 | 7.77 |
| | +AvgPool(ratio=1) | 47.06 | 76.97 | 85.61 | 2.0 | 9.53 | 62.09 | 83.58 | 89.40 | 1.0 | 7.73 |
| | +InvGC | 47.09 | 77.00 | 85.64 | 2.0 | **9.48** | **70.60** | **89.10** | **91.64** | 1.0 | **5.39** |
| | +InvGC w/LocalAdj | **47.47** | **77.46** | **85.84** | 2.0 | 9.53 | 66.42 | 85.37 | 90.00 | 1.0 | 7.37 |
| X-CLIP | | 46.31 | **76.84** | 85.31 | 2.0 | **9.59** | 65.52 | 83.73 | **89.85** | 1.0 | 8.15 |
| | +AvgPool(ratio=0.1) | 46.36 | 76.70 | 85.28 | 2.0 | 9.66 | 65.52 | 83.88 | 89.55 | 1.0 | 8.16 |
| | +AvgPool(ratio=0.25) | 46.33 | 76.71 | 85.28 | 2.0 | 9.64 | 65.37 | 84.03 | 89.55 | 1.0 | 8.17 |
| | +AvgPool(ratio=0.5) | 46.35 | 76.71 | 85.25 | 2.0 | 9.62 | 65.22 | 83.73 | 89.55 | 1.0 | 8.19 |
| | +AvgPool(ratio=1) | 46.41 | 76.77 | 85.27 | 2.0 | 9.60 | 65.37 | 83.88 | 89.55 | 1.0 | 8.16 |
| | +InvGC | **46.82** | 76.69 | **85.38** | 2.0 | 9.63 | **70.45** | 85.22 | 89.55 | 1.0 | 7.76 |
| | +InvGC w/LocalAdj | 46.49 | 76.82 | 85.29 | 2.0 | 9.63 | 66.27 | 84.33 | 89.25 | 1.0 | **7.74** |

Table 16: Retrieval performance on DiDeMo. Best in **Bold** and the second best is underlined.

| Method | Normalization | Text-to-Video Retrieval | | | | | Video-to-Text Retrieval | | | | |
|---|---|---|---|---|---|---|---|---|---|---|---|
| | | R@1↑ | R@5↑ | R@10↑ | MdR↓ | MnR↓ | R@1↑ | R@5↑ | R@10↑ | MdR↓ | MnR↓ |
| CE+ | | 18.23 | 42.63 | 56.08 | 8.0 | 42.05 | 18.82 | 42.73 | 55.78 | 8.0 | 37.27 |
| | +AvgPool(ratio=0.1) | 18.23 | 42.73 | 56.27 | 8.0 | 42.19 | 18.82 | 42.03 | 55.88 | 8.0 | 37.73 |
| | +AvgPool(ratio=0.25) | 18.23 | 42.73 | 56.27 | 8.0 | 42.11 | 18.92 | 42.43 | 55.88 | 8.0 | 37.36 |
| | +AvgPool(ratio=0.5) | 18.13 | 42.83 | 56.18 | 8.0 | 42.07 | 18.92 | 42.33 | 55.98 | 8.0 | 37.52 |
| | +AvgPool(ratio=1) | 18.23 | 42.63 | 56.08 | 8.0 | 42.05 | 18.82 | 42.73 | 55.78 | 8.0 | 37.27 |
| | +InvGC | 18.82 | **43.13** | **56.67** | 8.0 | **41.69** | 19.02 | 42.43 | 56.18 | 8.0 | **37.08** |
| | +InvGC w/LocalAdj | **19.02** | 43.03 | **56.67** | 8.0 | 42.04 | **19.42** | **43.53** | **56.70** | 8.0 | 39.28 |
| TT-CE+ | | 22.31 | 49.20 | 62.15 | 6.0 | 31.18 | 21.41 | 47.11 | 60.66 | 6.0 | 29.67 |
| | +AvgPool(ratio=0.1) | 22.31 | 49.50 | 62.85 | 6.0 | 31.18 | 21.41 | 47.21 | 60.46 | 6.0 | 29.74 |
| | +AvgPool(ratio=0.25) | 22.41 | 49.30 | 62.15 | 6.0 | 31.18 | 21.41 | 47.21 | 60.56 | 6.0 | 29.71 |
| | +AvgPool(ratio=0.5) | 22.41 | 49.30 | 62.15 | 6.0 | **31.17** | 21.41 | 47.21 | 60.66 | 6.0 | 29.69 |
| | +AvgPool(ratio=1) | 22.31 | 49.20 | 62.15 | 6.0 | 31.18 | 21.41 | 47.11 | 60.66 | 6.0 | 29.67 |
| | +InvGC | 23.80 | 50.10 | **63.75** | 5.25 | 31.51 | 22.21 | 47.81 | 61.35 | 6.0 | **29.53** |
| | +InvGC w/LocalAdj | **24.10** | 50.30 | 62.45 | **5.0** | 31.76 | **23.90** | 49.60 | 62.95 | 6.0 | 33.76 |

Table 17: Audio-text retrieval performance on CLOTHO. Best in **Bold** and the second best is underlined. '*' refers to the results direct copied from (Koepke et al., 2022). MoEE (Miech et al., 2020), MMT (Gabeur et al., 2020) are two methods used for audio-text retrieval.

| Method | Normalization | Text-to-Audio Retrieval | | | | | Audio-to-Text Retrieval | | | | |
|---|---|---|---|---|---|---|---|---|---|---|---|
| | | R@1↑ | R@5↑ | R@10↑ | MdR↓ | MnR↓ | R@1↑ | R@5↑ | R@10↑ | MdR↓ | MnR↓ |
| MoEE* | | 6.00 | 20.80 | 32.30 | 23.0 | 60.20 | 7.20 | 22.10 | 33.20 | 22.7 | 71.80 |
| MMT* | | 6.50 | 21.60 | 66.90 | 23.0 | 67.70 | 6.30 | 22.80 | 33.30 | 22.3 | 67.30 |
| AR-CE | | 6.27 | 22.32 | 33.30 | 23.0 | 58.96 | 7.27 | 23.35 | 35.12 | **21.0** | 74.68 |
| | +AVGPOOL(ratio=0.1) | 6.37 | 22.33 | 33.59 | 23.0 | 65.57 | 7.66 | 22.01 | 33.59 | 22.0 | 85.61 |
| | +AVGPOOL(ratio=0.25) | 6.47 | 22.12 | 33.58 | 22.0 | 59.84 | 7.85 | 22.78 | 34.64 | **21.0** | 76.21 |
| | +AVGPOOL(ratio=0.5) | 6.39 | 22.24 | 33.30 | 22.0 | 59.14 | 7.56 | 22.87 | 34.74 | **21.0** | 74.91 |
| | +AVGPOOL(ratio=1) | 6.28 | 22.32 | 33.30 | 23.0 | 58.95 | 7.27 | 23.35 | 35.12 | **21.0** | 74.68 |
| | +INVGC | **6.81** | 22.14 | **34.72** | **21.0** | 61.98 | **9.19** | 24.02 | 35.41 | 22.0 | 75.06 |
| | +INVGC w/LOCALADJ | 6.58 | **22.35** | 34.64 | 22.0 | **58.51** | 9.00 | **24.50** | **36.17** | **21.0** | **71.69** |

Table 18: Text-audio retrieval performance on AudioCaps. Best in **Bold** and the second best is underlined. '*' refers to the results direct copied from (Koepke et al., 2022). MoEE (Miech et al., 2020), MMT (Gabeur et al., 2020) are two methods used for audio-text retrieval.

| Method | Normalization | Text-to-Audio Retrieval | | | | | Audio-to-Text Retrieval | | | | |
|---|---|---|---|---|---|---|---|---|---|---|---|
| | | R@1↑ | R@5↑ | R@10↑ | MdR↓ | MnR↓ | R@1↑ | R@5↑ | R@10↑ | MdR↓ | MnR↓ |
| MoEE* | | 23.00 | 55.70 | 71.00 | 4.0 | 16.30 | 26.60 | 59.30 | 73.50 | 4.0 | 15.60 |
| MMT* | | 36.10 | 72.00 | 84.50 | 2.3 | 7.50 | 39.60 | 76.80 | 86.70 | 2.0 | 6.50 |
| AR-CE | | 22.33 | 54.49 | 70.54 | 5.0 | **15.89** | 24.02 | 56.00 | 71.81 | 4.0 | 16.91 |
| | +AVGPOOL(ratio=0.1) | 22.33 | 54.49 | 70.54 | 5.0 | **15.89** | 24.14 | 55.88 | 71.69 | 4.0 | 16.93 |
| | +AVGPOOL(ratio=0.25) | 22.33 | 54.49 | 70.54 | 5.0 | **15.89** | 24.14 | 56.13 | 71.69 | 4.0 | 16.92 |
| | +AVGPOOL(ratio=0.5) | 22.33 | 54.49 | 70.54 | 5.0 | **15.89** | 24.14 | 56.13 | 71.69 | 4.0 | 16.92 |
| | +AVGPOOL(ratio=1) | 22.33 | 54.49 | 70.54 | 5.0 | **15.89** | 24.02 | 56.00 | 71.81 | 4.0 | 16.91 |
| | +INVGC | 22.33 | 54.46 | **70.56** | 5.0 | **15.89** | 24.88 | 58.21 | 71.08 | 4.0 | 17.18 |
| | +INVGC w/LOCALADJ | **24.07** | **55.69** | 70.20 | **4.0** | 16.54 | **27.57** | **59.31** | **74.75** | 4.0 | **14.21** |

Table 19: Similarity measures within gallery set (MeanSim@1 equivalent to $\Delta_{deg}(G)$) on CLOTHO.

| Method | Normalization | Text-to-Audio Retrieval | | | Audio-to-Text Retrieval | | |
|---|---|---|---|---|---|---|---|
| | | MeanSim | MeanSim@1 | MeanSim@10 | MeanSim | MeanSim@1 | MeanSim@10 |
| AR-CE | | 0.0436 | 0.7518 | 0.6652 | 0.0143 | 0.2462 | 0.2144 |
| | +INVGC | 0.0391 | 0.5246 | 0.4567 | 0.0115 | 0.1261 | 0.1017 |
| | +INVGC w/LOCALADJ | 0.0279 | 0.4739 | 0.4094 | 0.0114 | 0.1713 | 0.1420 |

Table 20: Similarity measures within gallery set (MeanSim@1 equivalent to $\Delta_{deg}(G)$) on AudioCaps.

| Method | Normalization | Text-to-Audio Retrieval | | | Audio-to-Text Retrieval | | |
|---|---|---|---|---|---|---|---|
| | | MeanSim | MeanSim@1 | MeanSim@10 | MeanSim | MeanSim@1 | MeanSim@10 |
| AR-CE | | 0.0358 | 0.6781 | 0.5622 | 0.0075 | 0.2656 | 0.2330 |
| | +INVGC | 0.0358 | 0.6778 | 0.5620 | 0.006 | 0.1837 | 0.1582 |
| | +INVGC w/LOCALADJ | 0.0161 | 0.3264 | 0.2602 | 0.0054 | 0.1399 | 0.1142 |

Table 21: Similarity measures within gallery set (MeanSim@1 equivalent to $\Delta_{deg}(G)$) on MSCOCO (5k split).

| Method | Normalization | Text-to-Image Retrieval | | | Image-to-Text Retrieval | | |
|---|---|---|---|---|---|---|---|
| | | MeanSim | MeanSim@1 | MeanSim@10 | MeanSim | MeanSim@1 | MeanSim@10 |
| CLIP | | 0.4717 | 0.8211 | 0.7803 | 0.5161 | 0.9029 | 0.8596 |
| | +INVGC | 0.4693 | 0.8142 | 0.7738 | 0.5137 | 0.8972 | 0.8541 |
| | +INVGC w/LOCALADJ | 0.4646 | 0.8059 | 0.7647 | 0.5105 | 0.8924 | 0.8477 |
| Oscar | | 0.0509 | 0.6014 | 0.4867 | 0.0166 | 0.7894 | 0.6601 |
| | +INVGC | 0.0536 | 0.5960 | 0.4810 | 0.0172 | 0.7836 | 0.6523 |
| | +INVGC w/LOCALADJ | 0.0703 | 0.5610 | 0.4459 | 0.0374 | 0.7658 | 0.6285 |

Table 22: Similarity measures within gallery set (MeanSim@1 equivalent to $\Delta_{deg}(G)$) on Flickr30k.

| Method | Normalization | Text-to-Image Retrieval | | | Image-to-Text Retrieval | | |
|---|---|---|---|---|---|---|---|
| | | MeanSim | MeanSim@1 | MeanSim@10 | MeanSim | MeanSim@1 | MeanSim@10 |
| CLIP | | 0.4908 | 0.7639 | 0.7115 | 0.5119 | 0.8570 | 0.7882 |
| | +INVGC | 0.4868 | 0.7575 | 0.7056 | 0.5086 | 0.8515 | 0.7829 |
| | +INVGC w/LOCALADJ | 0.4671 | 0.7481 | 0.6942 | 0.4682 | 0.8358 | 0.7604 |
| Oscar | | 0.0143 | 0.4155 | 0.3007 | 0.0102 | 0.7059 | 0.5050 |
| | +INVGC | 0.0147 | 0.4015 | 0.2854 | 0.0116 | 0.6819 | 0.4778 |
| | +INVGC w/LOCALADJ | 0.0165 | 0.4077 | 0.2948 | 0.0139 | 0.6915 | 0.4880 |

Table 23: Similarity measures within gallery set (MeanSim@1 equivalent to $\Delta_{deg}(G)$) on MSR-VTT (full split and 1k split).

| Method | Normalization | Text-to-Video Retrieval | | | Video-to-Text Retrieval | | |
|---|---|---|---|---|---|---|---|
| | | MeanSim | MeanSim@1 | MeanSim@10 | MeanSim | MeanSim@1 | MeanSim@10 |
| | | MSR-VTT (full split) | | | | | |
| CE+ | | 0.1525 | 0.6750 | 0.5516 | 0.0006 | 0.0250 | 0.0223 |
| | +INVGC | 0.1435 | 0.5974 | 0.4861 | 0.0005 | 0.0224 | 0.0199 |
| | +INVGC w/LOCALADJ | 0.0845 | 0.4722 | 0.3697 | 0.0007 | 0.0196 | 0.0171 |
| TT-CE+ | | 0.1751 | 0.6878 | 0.5696 | 0.0008 | 0.0295 | 0.0267 |
| | +INVGC | 0.1627 | 0.6080 | 0.5017 | 0.0007 | 0.0241 | 0.0216 |
| | +INVGC w/LOCALADJ | 0.1196 | 0.5176 | 0.4161 | 0.0007 | 0.0237 | 0.0212 |
| | | MSR-VTT (1k split) | | | | | |
| CLIP4Clip | | 0.5416 | 0.8201 | 0.7628 | 0.6073 | 0.8389 | 0.7922 |
| | +INVGC | 0.5460 | 0.8195 | 0.7618 | 0.6149 | 0.8403 | 0.7919 |
| | +INVGC w/LOCALADJ | 0.5280 | 0.8098 | 0.7502 | 0.6090 | 0.8385 | 0.7921 |
| CLIP2Video | | 0.5376 | 0.8179 | 0.7589 | 0.5769 | 0.8210 | 0.7702 |
| | +INVGC | 0.5379 | 0.8179 | 0.7588 | 0.5798 | 0.8216 | 0.7703 |
| | +INVGC w/LOCALADJ | 0.5314 | 0.8129 | 0.7528 | 0.5627 | 0.8077 | 0.7559 |
| X-CLIP | | 0.3555 | 0.7484 | 0.6675 | 0.2626 | 0.5262 | 0.4608 |
| | +INVGC | 0.3671 | 0.7367 | 0.6520 | 0.2653 | 0.5251 | 0.4600 |
| | +INVGC w/LOCALADJ | 0.3054 | 0.6974 | 0.6065 | 0.2860 | 0.5344 | 0.4710 |

Table 24: Similarity measures within gallery set (MeanSim@1 equivalent to $\Delta_{deg}(G)$) on ActivityNet.

| Method | Normalization | Text-to-Video Retrieval | | | Video-to-Text Retrieval | | |
|---|---|---|---|---|---|---|---|
| | | MeanSim | MeanSim@1 | MeanSim@10 | MeanSim | MeanSim@1 | MeanSim@10 |
| CE+ | | 0.1245 | 0.6950 | 0.6095 | 0.0012 | 0.0164 | 0.0141 |
| | +INVGC | 0.1177 | 0.6223 | 0.5453 | 0.0011 | 0.0161 | 0.0139 |
| | +INVGC w/LOCALADJ | 0.0601 | 0.4059 | 0.3489 | 0.0010 | 0.0129 | 0.0110 |
| TT-CE+ | | 0.1347 | 0.7502 | 0.6673 | 0.0021 | 0.0209 | 0.0187 |
| | +INVGC | 0.1329 | 0.7213 | 0.6412 | 0.0021 | 0.0206 | 0.0185 |
| | +INVGC w/LOCALADJ | 0.0570 | 0.4082 | 0.3563 | 0.0018 | 0.0145 | 0.0128 |
| CLIP4Clip | | 0.3411 | 0.8122 | 0.7566 | 0.2782 | 0.7638 | 0.6932 |
| | +INVGC | 0.3412 | 0.8122 | 0.7566 | 0.2795 | 0.7635 | 0.6929 |
| | +INVGC w/LOCALADJ | 0.3554 | 0.8029 | 0.7469 | 0.2800 | 0.7483 | 0.6754 |
| X-CLIP | | 0.3229 | 0.8097 | 0.7460 | 0.1842 | 0.6635 | 0.5816 |
| | +INVGC | 0.3369 | 0.8052 | 0.7410 | 0.1867 | 0.6553 | 0.5743 |
| | +INVGC w/LOCALADJ | 0.3033 | 0.7638 | 0.6926 | 0.1968 | 0.6464 | 0.5646 |

formance on the retrieval task from text to other modalities, which is the more important task in practice compared to the other direction.

Based on the results, we can safely draw the conclusion that both proposed methods are able to significantly alleviate the representation degeneration problem by significantly reducing $\Delta_{deg}(G)$ score.

**Continuation on RQ2: Is INVGC w/LOCALADJ sensitive to the hyperparameter**

Table 25: Similarity measures within gallery set (MeanSim@1 equivalent to $\Delta_{deg}(G)$) on MSVD.

| Method | Normalization | Text-to-Video Retrieval | | | Video-to-Text Retrieval | | |
| | | MeanSim | MeanSim@1 | MeanSim@10 | MeanSim | MeanSim@1 | MeanSim@10 |
|---|---|---|---|---|---|---|---|
| CE+ | | 0.1057 | 0.7050 | 0.5843 | 0.0155 | 0.0659 | 0.0641 |
| | +INVGC | 0.0464 | 0.4200 | 0.3249 | 0.0075 | 0.0548 | 0.0534 |
| | +INVGC w/LOCALADJ | 0.0491 | 0.3884 | 0.3046 | 0.0064 | 0.0327 | 0.0310 |
| TT-CE+ | | 0.1176 | 0.7156 | 0.5968 | 0.0193 | 0.0758 | 0.0739 |
| | +INVGC | 0.0493 | 0.4194 | 0.3228 | 0.0152 | 0.0565 | 0.0547 |
| | +INVGC w/LOCALADJ | 0.0478 | 0.3661 | 0.2864 | 0.0093 | 0.0415 | 0.0398 |
| CLIP4Clip | | 0.6790 | 0.8916 | 0.8518 | 0.6485 | 0.9736 | 0.9527 |
| | +INVGC | 0.6918 | 0.8896 | 0.8479 | 0.6505 | 0.9737 | 0.9528 |
| | +INVGC w/LOCALADJ | 0.6731 | 0.8838 | 0.8429 | 0.6481 | 0.9734 | 0.9522 |
| CLIP2Video | | 0.5376 | 0.8179 | 0.7589 | 0.5769 | 0.8210 | 0.7702 |
| | +INVGC | 0.5379 | 0.8179 | 0.7588 | 0.5798 | 0.8216 | 0.7703 |
| | +INVGC w/LOCALADJ | 0.5314 | 0.8129 | 0.7528 | 0.5627 | 0.8077 | 0.7559 |
| X-CLIP | | 0.6931 | 0.8961 | 0.8587 | 0.5796 | 0.9165 | 0.8945 |
| | +INVGC | 0.7035 | 0.8904 | 0.8491 | 0.5879 | 0.9159 | 0.8942 |
| | +INVGC w/LOCALADJ | 0.6819 | 0.8885 | 0.8492 | 0.5600 | 0.9163 | 0.8913 |

Table 26: Similarity measures within gallery set (MeanSim@1 equivalent to $\Delta_{deg}(G)$) on DiDeMo.

| Method | Normalization | Text-to-Video Retrieval | | | Video-to-Text Retrieval | | |
| | | MeanSim | MeanSim@1 | MeanSim@10 | MeanSim | MeanSim@1 | MeanSim@10 |
|---|---|---|---|---|---|---|---|
| CE+ | | 0.1375 | 0.6183 | 0.4835 | 0.0021 | 0.0176 | 0.0144 |
| | +INVGC | 0.1277 | 0.5686 | 0.4409 | 0.0019 | 0.0167 | 0.0136 |
| | +INVGC w/LOCALADJ | 0.1374 | 0.6001 | 0.4690 | 0.0019 | 0.0150 | 0.0123 |
| TT-CE+ | | 0.1609 | 0.6794 | 0.5580 | 0.0018 | 0.0304 | 0.0248 |
| | +INVGC | 0.1432 | 0.5299 | 0.4292 | 0.0015 | 0.0244 | 0.0197 |
| | +INVGC w/LOCALADJ | 0.0598 | 0.3680 | 0.2806 | 0.0018 | 0.0199 | 0.0161 |

Table 27: Similarity measures between test gallery and query set on CLOTHO. The nearest neighbor is considered with respect to the gallery set.

| Method | Normalization | Text-to-Audio Retrieval | | | Audio-to-Text Retrieval | | |
| | | MeanSim | MeanSim@1 | MeanSim@10 | MeanSim | MeanSim@1 | MeanSim@10 |
|---|---|---|---|---|---|---|---|
| AR-CE | | 0.0210 | 0.6028 | 0.5683 | 0.0210 | 0.5788 | 0.5212 |
| | +INVGC | 0.0195 | 0.3931 | 0.3657 | 0.0185 | 0.3235 | 0.2830 |
| | +INVGC w/LOCALADJ | 0.0146 | 0.4520 | 0.4231 | 0.0104 | 0.4044 | 0.3600 |

Table 28: Similarity measures between test gallery and query set on AudioCaps. The nearest neighbor is considered with respect to the gallery set.

| Method | Normalization | Text-to-Audio Retrieval | | | Audio-to-Text Retrieval | | |
| | | MeanSim | MeanSim@1 | MeanSim@10 | MeanSim | MeanSim@1 | MeanSim@10 |
|---|---|---|---|---|---|---|---|
| AR-CE | | 0.0036 | 0.6229 | 0.5729 | 0.0036 | 0.5927 | 0.4883 |
| | +INVGC | 0.0036 | 0.6228 | 0.5728 | 0.0037 | 0.4454 | 0.3595 |
| | +INVGC w/LOCALADJ | 0.0005 | 0.4058 | 0.3672 | -0.0068 | 0.3568 | 0.2882 |

$r_g$ **and** $r_q$**?**

To address the question, like analysis on INVGC, we assess the R@1 metrics of our proposed method under a wide range of hyperparameters compared to the optimal choice adopted. For each task, one of $r_g$ or $r_q$ is fixed and the other is tuned to observe its impact on the R@1 metrics. The results obtained from the MSCOCO dataset are depicted in Figure 6. Despite some observed variations, our method consistently outperforms the baseline, represented by the red dashed line. This suggests that the proposed method can continuously enhance performance even when parameters are varied over a broad range.

Table 29: Similarity measures between test gallery and query set on MSCOCO (5k split). The nearest neighbor is considered with respect to the gallery set.

| Method | Normalization | Text-to-Image Retrieval | | | Image-to-Text Retrieval | | |
|---|---|---|---|---|---|---|---|
| | | MeanSim | MeanSim@1 | MeanSim@10 | MeanSim | MeanSim@1 | MeanSim@10 |
| CLIP | | 0.1516 | 0.3282 | 0.3138 | 0.1516 | 0.3213 | 0.3009 |
| | +INVGC | 0.1500 | 0.3245 | 0.3101 | 0.1511 | 0.3198 | 0.2994 |
| | +INVGC w/LOCALADJ | 0.0635 | 0.2214 | 0.2035 | 0.0921 | 0.2414 | 0.2208 |
| Oscar | | -0.0188 | 0.5423 | 0.4790 | -0.0188 | 0.4893 | 0.3940 |
| | +INVGC | -0.0192 | 0.5379 | 0.4744 | -0.0193 | 0.4827 | 0.3873 |
| | +INVGC w/LOCALADJ | -0.0257 | 0.4815 | 0.4172 | -0.0368 | 0.4290 | 0.3361 |

Table 30: Similarity measures between test gallery and query set on Flickr30k. The nearest neighbor is considered with respect to the gallery set.

| Method | Normalization | Text-to-Image Retrieval | | | Image-to-Text Retrieval | | |
|---|---|---|---|---|---|---|---|
| | | MeanSim | MeanSim@1 | MeanSim@10 | MeanSim | MeanSim@1 | MeanSim@10 |
| CLIP | | 0.1688 | 0.3167 | 0.2980 | 0.1688 | 0.3030 | 0.2777 |
| | +INVGC | 0.1681 | 0.3156 | 0.2969 | 0.1683 | 0.3020 | 0.2769 |
| | +INVGC w/LOCALADJ | 0.0854 | 0.2539 | 0.2295 | 0.1150 | 0.2440 | 0.2190 |
| Oscar | | -0.0073 | 0.4349 | 0.3552 | -0.0073 | 0.3738 | 0.2695 |
| | +INVGC | -0.0074 | 0.4235 | 0.3440 | -0.0078 | 0.3441 | 0.2403 |
| | +INVGC w/LOCALADJ | -0.0084 | 0.4259 | 0.3471 | -0.0102 | 0.3562 | 0.2554 |

Table 31: Similarity measures between test gallery and query set on MSR-VTT (full split and 1k split). The nearest neighbor is considered with respect to the gallery set.

| Method | Normalization | Text-to-Video Retrieval | | | Video-to-Text Retrieval | | |
|---|---|---|---|---|---|---|---|
| | | MeanSim | MeanSim@1 | MeanSim@10 | MeanSim | MeanSim@1 | MeanSim@10 |
| MSR-VTT (full split) | | | | | | | |
| CE+ | | -0.0026 | 0.4262 | 0.4021 | -0.0026 | 0.3830 | 0.3367 |
| | +INVGC | -0.0024 | 0.3906 | 0.3680 | -0.0025 | 0.3474 | 0.3047 |
| | +INVGC w/LOCALADJ | -0.0022 | 0.3610 | 0.3395 | -0.0142 | 0.2680 | 0.2316 |
| TT-CE+ | | 0.0135 | 0.4632 | 0.4404 | 0.0135 | 0.4245 | 0.3774 |
| | +INVGC | 0.0130 | 0.4234 | 0.4023 | 0.0123 | 0.3533 | 0.3128 |
| | +INVGC w/LOCALADJ | 0.0103 | 0.3779 | 0.3578 | 0.0028 | 0.3273 | 0.2880 |
| MSR-VTT (1k split) | | | | | | | |
| CLIP4Clip | | 0.1074 | 0.2399 | 0.2160 | 0.1074 | 0.2388 | 0.2131 |
| | +INVGC | 0.1078 | 0.2397 | 0.2159 | 0.1079 | 0.2394 | 0.2137 |
| | +INVGC w/LOCALADJ | 0.1061 | 0.2394 | 0.2149 | 0.0906 | 0.2176 | 0.1923 |
| CLIP2Video | | 0.1214 | 0.2689 | 0.2344 | 0.1214 | 0.2686 | 0.2325 |
| | +INVGC | 0.1214 | 0.2688 | 0.2343 | 0.1217 | 0.2687 | 0.2327 |
| | +INVGC w/LOCALADJ | 0.1153 | 0.2629 | 0.2278 | 0.0502 | 0.1825 | 0.1470 |
| X-CLIP | | 0.1241 | 0.2484 | 0.2209 | 0.1241 | 0.2459 | 0.2158 |
| | +INVGC | 0.1248 | 0.2479 | 0.2199 | 0.1241 | 0.2442 | 0.2142 |
| | +INVGC w/LOCALADJ | 0.1197 | 0.2441 | 0.2154 | 0.0895 | 0.1955 | 0.1669 |

# D  Proofs

## D.1  Proof of Theorem 1

To prove the theorem, we start with some preliminary definitions. We denote the $n$-dimensional unit sphere as,

$$\mathcal{S}_{n-1} := \{\mathbf{x} \in \mathbb{R}^n : \|\mathbf{x}\|_2 = 1\}$$

and $n$-dimensional ball,

$$\mathcal{B}_{n,b} := \{\mathbf{x} \in \mathbb{R}^n : \|x\|_2 \leq a\}$$

Then, $n$-dimensional ball representing the neighborhood of $\mathbf{x}$ with radius $a$ can be denoted as $\mathcal{B}_{n,\mathbf{x},a} := \{\mathbf{y} \in \mathbb{R}^n : \|y - x\|_2 \leq a\}$. We go on to denote spherical caps $\mathbf{C}_{n,\mathbf{x},a}$ as

$$\mathbf{C}_{n,\mathbf{x},b} := \mathcal{S}_{n-1} \cap \mathcal{B}_{n,\mathbf{x},b}$$

Table 32: Similarity measures between test gallery and query set on ActivityNet. The nearest neighbor is considered with respect to the gallery set.

| Method | Normalization | Text-to-Video Retrieval | | | Video-to-Text Retrieval | | |
|---|---|---|---|---|---|---|---|
| | | MeanSim | MeanSim@1 | MeanSim@10 | MeanSim | MeanSim@1 | MeanSim@10 |
| CE+ | | 0.0210 | 0.4209 | 0.3703 | 0.0210 | 0.4154 | 0.3679 |
| | +INVGC | 0.0204 | 0.4066 | 0.3578 | 0.0208 | 0.4090 | 0.3623 |
| | +INVGC w/LOCALADJ | 0.0149 | 0.3304 | 0.2893 | -0.0017 | 0.3049 | 0.2673 |
| TT-CE+ | | 0.0391 | 0.5070 | 0.4611 | 0.0391 | 0.5019 | 0.4591 |
| | +INVGC | 0.0383 | 0.4885 | 0.4440 | 0.0389 | 0.4973 | 0.4550 |
| | +INVGC w/LOCALADJ | 0.0258 | 0.3796 | 0.3429 | 0.0148 | 0.3330 | 0.3017 |
| CLIP4Clip | | 0.0533 | 0.3567 | 0.3223 | 0.0533 | 0.3567 | 0.3220 |
| | +INVGC | 0.0533 | 0.3567 | 0.3222 | 0.0533 | 0.3553 | 0.3206 |
| | +INVGC w/LOCALADJ | 0.0106 | 0.2821 | 0.2488 | 0.0107 | 0.2854 | 0.2512 |
| X-CLIP | | 0.1233 | 0.3484 | 0.3135 | 0.1233 | 0.3464 | 0.3122 |
| | +INVGC | 0.1233 | 0.3460 | 0.3111 | 0.1219 | 0.3397 | 0.3058 |
| | +INVGC w/LOCALADJ | 0.1089 | 0.3018 | 0.2683 | 0.0968 | 0.3005 | 0.2672 |

Table 33: Similarity measures between test gallery and query set on MSVD. The nearest neighbor is considered with respect to the gallery set.

| Method | Normalization | Text-to-Video Retrieval | | | Video-to-Text Retrieval | | |
|---|---|---|---|---|---|---|---|
| | | MeanSim | MeanSim@1 | MeanSim@10 | MeanSim | MeanSim@1 | MeanSim@10 |
| CE+ | | 0.1029 | 0.6456 | 0.6188 | 0.1029 | 0.6037 | 0.5235 |
| | +INVGC | 0.0159 | 0.4832 | 0.4624 | 0.0358 | 0.3979 | 0.3408 |
| | +INVGC w/LOCALADJ | 0.0450 | 0.4501 | 0.4279 | 0.0384 | 0.3393 | 0.2870 |
| TT-CE+ | | 0.1170 | 0.6681 | 0.6417 | 0.1170 | 0.5997 | 0.531 |
| | +INVGC | 0.0613 | 0.5144 | 0.4924 | 0.1019 | 0.5075 | 0.4456 |
| | +INVGC w/LOCALADJ | 0.0421 | 0.4581 | 0.4369 | 0.0574 | 0.3807 | 0.3347 |
| CLIP4Clip | | 0.1273 | 0.2686 | 0.2606 | 0.1273 | 0.2483 | 0.2242 |
| | +INVGC | 0.1283 | 0.2674 | 0.2593 | 0.1273 | 0.2473 | 0.2233 |
| | +INVGC w/LOCALADJ | 0.0635 | 0.1982 | 0.1887 | 0.1167 | 0.2359 | 0.2119 |
| CLIP2Video | | 0.1413 | 0.3032 | 0.2931 | 0.1413 | 0.2779 | 0.2500 |
| | +INVGC | 0.1417 | 0.3031 | 0.2930 | 0.1416 | 0.2771 | 0.2492 |
| | +INVGC w/LOCALADJ | 0.1073 | 0.2635 | 0.2526 | 0.0770 | 0.1992 | 0.1723 |
| X-CLIP | | 0.1672 | 0.2943 | 0.2858 | 0.1672 | 0.2728 | 0.2487 |
| | +INVGC | 0.1678 | 0.2940 | 0.2853 | 0.1675 | 0.2722 | 0.2482 |
| | +INVGC w/LOCALADJ | 0.1621 | 0.2893 | 0.2805 | 0.1518 | 0.2564 | 0.2320 |

Table 34: Similarity measures between test gallery and query set on DiDeMo. The nearest neighbor is considered with respect to the gallery set.

| Method | Normalization | Text-to-Video Retrieval | | | Video-to-Text Retrieval | | |
|---|---|---|---|---|---|---|---|
| | | MeanSim | MeanSim@1 | MeanSim@10 | MeanSim | MeanSim@1 | MeanSim@10 |
| CE+ | | -0.0068 | 0.2960 | 0.2379 | -0.0068 | 0.2888 | 0.2369 |
| | +INVGC | -0.0066 | 0.2824 | 0.2265 | -0.0067 | 0.2825 | 0.2316 |
| | +INVGC w/LOCALADJ | -0.0135 | 0.2715 | 0.2165 | -0.0148 | 0.2338 | 0.1897 |
| TT-CE+ | | -0.002 | 0.4218 | 0.3485 | -0.002 | 0.4067 | 0.3439 |
| | +INVGC | -0.0007 | 0.3374 | 0.2755 | -0.0012 | 0.3747 | 0.3162 |
| | +INVGC w/LOCALADJ | -0.0082 | 0.2832 | 0.2302 | -0.019 | 0.2396 | 0.1983 |

Note that the radius $a$ here is not the neighborhood coverage $b$, where they have a relation as $a^2 = 2\sqrt{a^2 - b^2}$ by simple triangle calculation. Since the basis of $\mathbf{C}_{R,\mathbf{x},a}$ is an $n-1$ dimensional hypersphere of radius $b$ and $b$ is the key factor we are interested in, we denote the spherical caps using $b$ as,

$$\mathrm{Cap}_{n,\mathbf{x},b} := \mathbf{C}_{n,\mathbf{x},a}$$

Since the cosine similarity applied is indepen-

dent of the norm of each point in $G$ and $Q$, we can assume without loss of generality that all the points in $G$ and $Q$ are on the unit sphere, i.e. $\forall \mathbf{x} \in G \cup Q, x \in \mathcal{S}_{n-1}$. Since query data follows an independent and identical uniform distribution in $\mathbb{R}^n$, therefore it still follows an independent and identical uniform distribution on $\mathcal{S}_{n-1}$.

Note that the probability of $\mathbf{x_1}$ is correctly retrieved is lower bounded by the probability that the corresponding query point $\mathbf{q} \in \mathrm{Cap}_{n,\mathbf{x_1},b}$, for $q$ must be the nearest neighbor of $\mathbf{x_1}$.

Given uniform distribution, the probability is $\frac{\mathrm{A}(\mathrm{Cap}_{n,\mathbf{x_1},b})}{\mathrm{A}(\mathcal{S}_{n-1})}$, where the A is the operator for area of hypersurface.

Generally, bounding volume is easier than the surface area, so we have the following Lemma Lemma 3 to help establish the relationship between the two.

**Lemma 3.** *Therefore, the relationship between the surface area* $\mathrm{A}(\mathrm{Cap}_{n,\mathbf{x},b})$ *and volumn* $\mathrm{V}(\mathrm{Cap}_{n,\mathbf{x},b})$[5] *of n-dimensional sphere cap is,*

$$\mathrm{A}(\mathrm{Cap}_{n,\mathbf{x},b}) = n \, \mathrm{V}(\mathrm{Cap}_{n,\mathbf{x},b}) \qquad (12)$$

*Proof.* We have the relationship between the surface area $A(\mathcal{S}_{n-1})$ and volume $V(\mathcal{S}_{n-1})$ [6] of $n$-dimensional unit hypersphere as,

$$\mathrm{A}(S_{n-1}) = \frac{d}{dr} \, \mathrm{V}(S_{n-1}) = n \, \mathrm{V}(S_{n-1}) \quad (13)$$

Therefore, the same relationship can be adapted to sphere cap as well. $\qquad \square$

With Lemma 3, we only need to bound the volume $\mathrm{V}(\mathrm{Cap}_{n,\mathbf{x},b})$, which is given by the following Theorem 4.

**Theorem 4.** *Given n-dimensional unit sphere $\mathcal{S}_{n-1}$ and spherical caps $\mathrm{Cap}_{n,\mathbf{x},b}$ on it, we have,*

$$\frac{1}{2} \cdot b^n > \frac{\mathrm{V}(\mathrm{Cap}_{n,\mathbf{x},b})}{\mathrm{V}(\mathcal{S}_{n-1})} > \frac{1}{4n} \cdot b^{n+1}.$$

*Proof.* **Lower bound**: Follow the proof of Lemma 4.1 in (Micciancio and Voulgaris, 2010). The basis

---

[5]Every time we mention the hypervolume of a sphere $\mathrm{V}(\mathrm{Cap}_{n,\mathbf{x},b})$ where $\mathrm{Cap}_{n,\mathbf{x},b} = \mathbf{C}_{n,\mathbf{x},a} = \mathcal{S}_{n-1} \cap \mathcal{B}_{n,\mathbf{x},a}$, we actually always refer to that of $\mathrm{V}(\mathcal{B}_{n,1} \cap \mathcal{B}_{n,\mathbf{x},a})$. We keep using expressions like $\mathrm{V}(\mathrm{Cap}_{n,\mathbf{x},b})$ to prevent possible confusion when dealing with the area and volume at the same time

[6]When mentioning the hypervolume of a sphere $\mathcal{S}_{n-1}$, we actually always refer to that of $\mathcal{B}_{n,1}$.

of $\mathrm{Cap}_{R,\mathbf{x},b}$ is an $n-1$ dimensional hypersphere of radius $r = b$, denoted as $\mathcal{B}_{n-1,b}$. Therefore $\mathrm{Cap}_{n,\mathbf{x},b}$ includes a cone $C_1$ with basis as $\mathcal{B}_{n-1,b}$ and height $h = 1 - \sqrt{1 - b^2}$. Then, a cylinder $C_2$ with the same basis but height $2 \cdot b$ includes $\mathcal{B}_{n,b}$. Note that we have $h > b^2/2$ by Taylor expansion. Based on all the facts, we have:

$$\mathrm{V}(\mathrm{Cap}_{n,\mathbf{x},b}) > \mathrm{V}(C_1) = \mathrm{V}(\mathcal{B}_{n-1,b}) \frac{h}{n}$$

$$= \mathrm{V}(C_2) \frac{h}{2bn}$$

$$> \mathrm{V}(\mathcal{B}_{n,b}) \frac{h}{2bn}$$

$$> \mathrm{V}(\mathcal{B}_{n,b}) \frac{b}{4n}$$

Therefore,

$$\frac{b}{4n} \cdot \frac{\mathrm{V}(\mathrm{Cap}_{n,\mathbf{x},b})}{\mathrm{V}(\mathcal{S}_{n-1})} > \frac{b}{4n} \cdot \frac{\mathrm{V}(\mathcal{B}_{n,b})}{\mathrm{V}(\mathcal{B}_{n,1})}$$

$$= \frac{b}{4n} \cdot \left(\frac{b}{1}\right)^n$$

$$= \frac{1}{4n} \cdot b^{n+1}$$

**Upper bound**: Based on proof of Lower bound, we notice that half of $\mathcal{B}_{n,b}$ includes $\mathrm{Cap}_{n,\mathbf{x},b}$, we have

$$\mathrm{V}(\mathrm{Cap}_{n,\mathbf{x},b}) < \frac{\mathrm{V}(\mathcal{B}_{n,b})}{2}$$

Therefore,

$$\frac{\mathrm{V}(\mathrm{Cap}_{n,\mathbf{x},b})}{\mathrm{V}(\mathcal{S}_{n-1})} < \frac{1}{2} \cdot \frac{\mathrm{V}(\mathcal{B}_{n,b})}{\mathrm{V}(\mathcal{B}_{n,1})}$$

$$= \frac{1}{2} \cdot b^n$$

$\qquad \square$

Finally, we finish the proof using Lemma 3.

### D.2 Proof of Corollary 2

*Proof.* Using the result from Theorem 1, we have the following inequalities for the probability of successful retrieval of $b_1$ and $b_2$,

$$\mathrm{P}(\mathbf{x}, b_1) < \frac{n}{2} \cdot b_1^n,$$

And,

$$\mathrm{P}(\mathbf{x}, b_2) > \frac{1}{4} \cdot b_2^{n+1},$$

Therefore, we have,

$$\frac{\mathrm{P}(\mathbf{x_1}, b_1)}{\mathrm{P}(\mathbf{x_1}, b_2)} < \frac{2n}{b_2} \cdot \left(\frac{b_1}{b_2}\right)^n .$$

Note that $2/b_2$ is constant with respect to $n$, so we finish the proof. $\square$

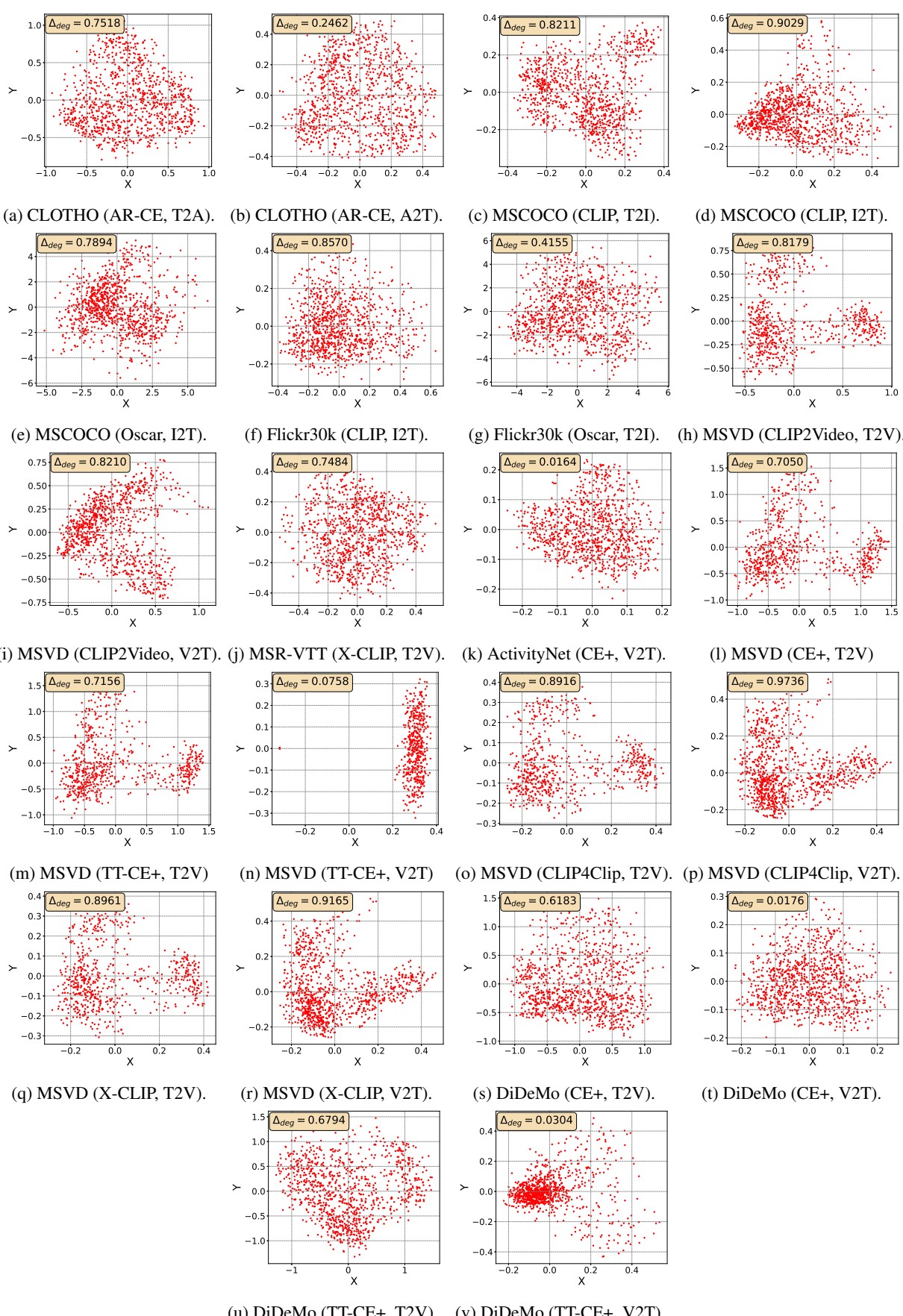

Figure 5: **The prevalence of representation degeneration problem across various methods and datasets**. Each figure illustrates the distribution of the gallery set's representation for both retrieval tasks. Dimension reduction is performed using PCA, and the first two principal dimensions are chosen.

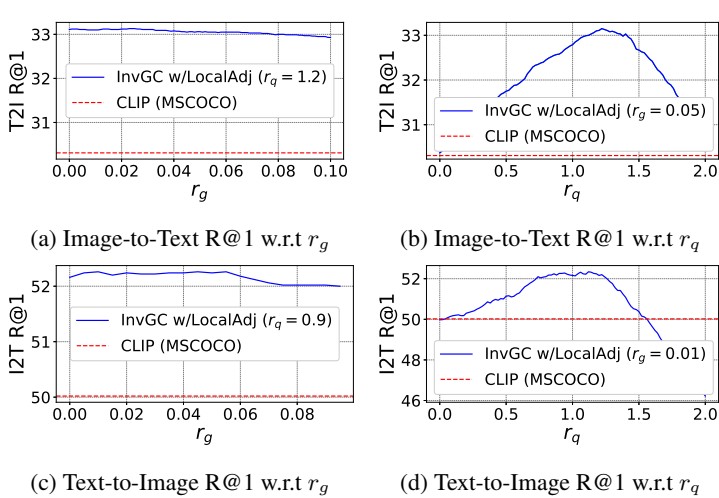

(a) Image-to-Text R@1 w.r.t $r_g$      (b) Image-to-Text R@1 w.r.t $r_q$

(c) Text-to-Image R@1 w.r.t $r_g$      (d) Text-to-Image R@1 w.r.t $r_q$

Figure 6: Hyperpapameters sensitivity of INVGC w/LOCALADJ on MSCOCO.