# OpenReview forum: "InvGC: Robust Cross-Modal Retrieval by Inverse Graph Convolution"
_EMNLP/2023/Conference — EMNLP 2023 Findings_

### Official Review · Reviewer_LxR3 · 2023-08-02

**Typos Grammar Style And Presentation Improvements:** n/a
**Soundness:** 3

**Excitement:**

4: Strong: This paper deepens the understanding of some phenomenon or lowers the barriers to an existing research direction.

**Missing References:**

	Some papers (e.g., Learning with Twin Noisy Labels for Visible-Infrared Person Re-Identification, Learning with Noisy Correspondence for Cross-modal Matching) on robust cross-modal retrieval topics are encouraged to be discussed.

**Paper Topic And Main Contributions:**

This paper studies the representation degeneration problem and reveals the influence of the problem for the cross-modal retrieval task. To solve the intractable problem, the authors propose a simple but effective method called INVGC, which effectively separates representations by increasing the distances between data points. Extensive empirical results validate the effectiveness of the proposed method.

**Questions For The Authors:**

	This paper seems to acquire some validation data. The practicality of the setting should be further clarified.
	The mathematical notations are confusing. For example, $n$ in Theorem 1 refers to the dimension of the representation while it refers to the data number in Eq. (2). I strongly recommend the authors to carefully revise all ambiguous notations for correctness.


**Reasons To Accept:**

	This paper extends the scope of the limited convex cone (as representation degeneration) problem into the cross-modal retrieval task, revealing the influence of this problem in the task. I think it is an interesting and practical problem for researchers in the community.
	The proposed method is simple but effectively alleviate the influence of representation degeneration problem for cross-modal retrieval task, as verified with the experiment results.
	Extensive empirical results validate the effectiveness of the proposed method.


**Reasons To Reject:**

	This paper seems to acquire some validation data. The practicality of the setting should be further clarified.
	The mathematical notations are confusing. For example, $n$ in Theorem 1 refers to the dimension of the representation while it refers to the data number in Eq. (2). I strongly recommend the authors to carefully revise all ambiguous notations for correctness.


**Reproducibility:**

4: Could mostly reproduce the results, but there may be some variation because of sample variance or minor variations in their interpretation of the protocol or method.

**Reviewer Confidence:**

5: Positive that my evaluation is correct. I read the paper very carefully and I am very familiar with related work.

---

> ### Author Rebuttal · Authors · 2023-08-26
>
> We sincerely thank you for your time, efforts, and your detailed and positive comments. Below please find the responses (R) to some specific questions (Q).
>
> $\textbf{Questions For The Authors}$:
>
> **Q(1) [Validation data is needed]:**
> >R: Yes, we do need validation data to select the hyperparameters, i.e., $r_g$ and $r_q$. For the dataset that does not have a validation set, we use a subset of the training set to determine the hyperparameters, which improves the practicality of the proposed methods. Similarly, in practice, researchers can split the training set into training and validation sets for selecting the hyperparameters.
>
> **Q(2) [The practicality of the method]:**
> >R: Thank you for your suggestion! We use a validation set, or a very small subset of the training set, to determine the hyperparameters, which is standard for a post-processing method. Also, we only have **2 (3)** hyperparameters used in  $\textbf{InvGC}$( $\textbf{InvGC}$ w/$\textbf{LocalAdj}$) for each task, and the performance of retrieval is not sensitive to all of them as shown in Figures 5 and 6 (Page 29). Detailed settings and implementations including benchmarks, methods, implementation of the proposed methods, and computational complexity are presented in Appendix C (Page15  - Page17).
>
> **Q(3) [Ambiguous notations]:**
> >R: Thank you very much for your suggestion! We will thoroughly and carefully revise the paper to avoid any confusion that might occur.
>
> $\textbf{Missing Reference}$:
> >Thank you for the suggestion!
> > - [1] proposed a novel and strong method, called DART, to tackle Twin Noise Labels in visible infrared person re-identification. Similar to our study, the problem is also related to the issue of data quality of cross-modal tasks but in a way of addressing noisy annotation and correspondence.
> > - [2] addressed a similar issue and proposed Noisy Correspondence Rectifier to learn with noisy correspondence.
> >
> >Though not dealing with the same problem as our study, both papers offered insightful ideas for tackling data-related problems and archiving robustness performance in cross-modal matching and retrieval, which serves as a pretty good reference to enrich our work.
> >
> > We will revise our paper carefully and add the discussion with these papers in our final version.
>
> We sincerely thank you for your thoughtful comments and are open to further discussion on these matters.
>
>
> [1] M. Yang, Z. Huang, P. Hu, T. Li, J. Lv and X. Peng. Learning with Twin Noisy Labels for Visible-Infrared Person Re-Identification. 2022 IEEE/CVF Conference on Computer Vision and Pattern Recognition (CVPR), New Orleans, LA, USA, 2022, pp. 14288-14297, doi: 10.1109/CVPR52688.2022.01391.
>
> [2] Huang, Zhenyu and Niu, Guocheng and Liu, Xiao and Ding, Wenbiao and Xiao, Xinyan and Wu, Hua and Peng, Xi. Learning with Noisy Correspondence for Cross-modal Matching. 35th Conference on Neural Information Processing Systems (NeurIPS 2021), page 29406--29419.

---

### Official Review · Reviewer_6sSa · 2023-08-03

**Soundness:** 3

**Excitement:**

3: Ambivalent: It has merits (e.g., it reports state-of-the-art results, the idea is nice), but there are key weaknesses (e.g., it describes incremental work), and it can significantly benefit from another round of revision. However, I won't object to accepting it if my co-reviewers champion it.

**Paper Topic And Main Contributions:**

The paper proposes a post-processing technique InvGC for cross-modal retrieval, which is inspired by graph convolution and aims to solve the representation degeneration problem caused by CLIP, i.e., the representations are gathered in a very narrow cone in the embedding space.

**Questions For The Authors:**

1. How to observe the representation degeneration problem of different datasets from Figure 4.
2. How to set the percent p.

**Reasons To Accept:**

1. This paper proposes a post processing method for cross modal retrieval, which can alleviate representation degeneration problem without any training process or additional data.
2. Several CLIP based models are improved by InvGC.

**Reasons To Reject:**

1. The theory in section 2.2 seems flawed. "A higher \delta_deg score leads to bad retrieval performance" is counterintuitive. I think when the degeneration issue occurs, not only the similarity between the closest data is higher, but also the similarities between other samples are higher. Therefore, only observing the similarity between neighboring samples is one-sided.
2. Why the adjacency matrix is computed between the test galary samples and training query/galary samples. What about the test query samples? Why are the presentations of test query samples not updated?

**Reproducibility:**

4: Could mostly reproduce the results, but there may be some variation because of sample variance or minor variations in their interpretation of the protocol or method.

**Reviewer Confidence:**

4: Quite sure. I tried to check the important points carefully. It's unlikely, though conceivable, that I missed something that should affect my ratings.

---

> ### Author Rebuttal · Authors · 2023-08-27
>
> We sincerely thank you for your time, efforts, and your detailed and positive comments. Below please find the responses (R) to some specific questions (Q).
>
> $\textbf{Question}$:
>
> **Q(1) [How to understand Figure 4]**:
> > R: Thank you for your question! The severity of the data degeneration problem is proportional to **the convexity of the cones** that the embedding of data is located in. We notice that a large score of data degeneration score (i.e. $\Delta_{deg}$) always leads to more concentrated data as shown in Figure 4. For different methods with the same dataset (same dataset and same retrieval task), e.g., Figures 4(s) and 4(u), Figures 4(p) and 4(r), more gathering in a convex cone (similar to a triangle in 2d space), the larger the score, which can be clearly observed in the figure. However, it is hard to compare the degeneration problem between datasets since their distributions are different.
>
>
> **Q(2) [How to set percent $p$]**:
> > R: Thank you for the nice catch. $p$ controls the reception field of $\textbf{AvgPool}$. We uniformly chose 4 values, as $[0.1, 0.25, 0.5, 1]$, to approximately find the best performance. In practice, the optimal value of $p$ can be selected via a validation set. However, we don’t recommend using $\textbf{AvgPool}$ (served as a baseline) since it is beaten by our proposed methods, $\textbf{InvGC}$, with a large margin based on experimental results.
>
> $\textbf{Reason to reject}$:
>
> **Q(1) [The theory in section 2.2]**:
> > R: We agree with you. It’s likely the case that both similarities are high when data degeneration is severe. In practice, we normally only need top-$k$ neighbors when using recommendation systems. To simplify the proof and have a clear presentation, we only consider the similarity between nearest neighbors. And we believe that our theoretical findings can be easily extended to the top-$k$ scenario.
>
>
> **Q(2) [Computation of adjacency matrix]**:
> > R: Thank you for the good catch. We do not compute the adjacency matrix of test queries for three reasons:
> > 1. [Pratical settings]: The query data in practice is very **unstable** since it comes separately and independently (in practice each query cannot see each other), which is exactly the opposite case compared to gallery data.
> > 2. [Theoretical analysis]: The margin of correct retrieval is proved to be dependent on the distribution of the gallery set according to Theorem 1. The large distance between the representation of nearest neighbors in the gallery set, the exponentially large probability of achieving correct.
> > 3. [Experimental results]: Considering the convolution of the test query set when computing the adjacency matrix will **double the number of hyperparameters** in a post-processing method, which is very undesirable if it’s not able to significantly improve the performance.
> We also add experiments to compare the performance of the **original setting** (only considering the convolution of the test gallery set) with the setting (both the convolution of the test query and gallery set, noted as **‘two-side’** here). Due to the limited space, we only present the Text-to-Image R@1 on MSCOCO and MSR-VTT. Results are presented in the following Tables 1 and 2.  For each task, all 4 hyperparameters of the two-side setting are tuned to achieve the best performance. Note that it’s very common that the optimal value of the hyperparameters for the query set is 0, with which the **two-side setting degrades to the original setting ($\textbf{InvGC}$)**. Such cases are noted as degraded after R@1.
> >
> >In fact, two-side (computing adjacency matrix for test query and gallery) degrades to one-side (only computing adjacency matrix for test query, $\textbf{InvGC}$) for all the following experiments. This experimental study supports the claims in the two reasons mentioned above that the inverse graph convolution of the gallery set is all that matters when computing the adjacency matrix. Doing so might hurt the practicality of the proposed methods by making them more time-consuming and unstable.
> >
> >Thank you for your suggestion! We will add a careful and complete discussion of this in the revised version of the paper.
>
> $\textbf{Table 1: Performance of two-side setting on MSCOCO}$
>
> | Method           | Setting                                            | R@1 of Text-to-Image  |
> |------------------|----------------------------------------------------|--------------------------------|
> | $\textbf{CLIP}$  |                                                    | 30.34                          |
> |                  | +$\textbf{InvGC}$                                  | 32.70                          |
> |                  | +$\textbf{InvGC}$ (two-side)                       | _32.70_ (degraded)             |
> |                  | +$\textbf{InvGC}$ w/$\textbf{LocalAdj}$            | 33.11                          |
> |                  | +$\textbf{InvGC}$ w/$\textbf{LocalAdj}$ (two-side) | _33.11_ (degraded)             |
> | $\textbf{Oscar}$ |                                                    | 52.50                          |
> |                  | +$\textbf{InvGC}$                                  | 52.63                          |
> |                  | +$\textbf{InvGC}$ (two-side)                       | _52.63_ (degraded)             |
> |                  | +$\textbf{InvGC}$ w/$\textbf{LocalAdj}$            | 52.93                          |
> |                  | +$\textbf{InvGC}$ w/$\textbf{LocalAdj}$ (two-side) | _52.93_ (degraded)             |
>
>
> $\textbf{Table 2: Performance of two-side setting on MSR-VTT}$
>
> | Method               | Setting                                            | R@1 of Text-to-Image  |
> |----------------------|----------------------------------------------------|--------------------------------|
> | $\textbf{CE+}$       |                                                    | 13.51                          |
> |                      | +$\textbf{InvGC}$                                  | 13.57                          |
> |                      | +$\textbf{InvGC}$ (two-side)                       | _13.57_ (degraded)             |
> |                      | +$\textbf{InvGC}$ w/$\textbf{LocalAdj}$            | 13.83                          |
> |                      | +$\textbf{InvGC}$ w/$\textbf{LocalAdj}$ (two-side) | _13.53_ (degraded)             |
> | $\textbf{TT-CE+}$    |                                                    | 14.51                          |
> |                      | +$\textbf{InvGC}$                                  | 14.62                          |
> |                      | +$\textbf{InvGC}$ (two-side)                       | _14.62_ (degraded)             |
> |                      | +$\textbf{InvGC}$ w/$\textbf{LocalAdj}$            | 15.08                          |
> |                      | +$\textbf{InvGC}$ w/$\textbf{LocalAdj}$ (two-side) | _15.08_ (degraded)             |
> | $\textbf{CLIP4CLIP}$ |                                                    | 44.10                          |
> |                      | +$\textbf{InvGC}$                                  | 44.40                          |
> |                      | +$\textbf{InvGC}$ (two-side)                       | _44.40_ (degraded)             |
> |                      | +$\textbf{InvGC}$ w/$\textbf{LocalAdj}$            | 44.40                          |
> |                      | +$\textbf{InvGC}$ w/$\textbf{LocalAdj}$ (two-side) | _44.40_ (degraded)             |
> | $\textbf{X-CLIP}$    |                                                    | 46.30                          |
> |                      | +$\textbf{InvGC}$                                  | 47.30                          |
> |                      | +$\textbf{InvGC}$ (two-side)                       | _47.30_ (degraded)             |
> |                      | +$\textbf{InvGC}$ w/$\textbf{LocalAdj}$            | 47.10                          |
> |                      | +$\textbf{InvGC}$ w/$\textbf{LocalAdj}$ (two-side) | _47.10_ (degraded)             |
>
>
> We sincerely thank you for your thoughtful comments and are open to further discussion on these matters.

---

### Official Review · Reviewer_2BeH · 2023-08-05

**Soundness:** 3

**Excitement:**

3: Ambivalent: It has merits (e.g., it reports state-of-the-art results, the idea is nice), but there are key weaknesses (e.g., it describes incremental work), and it can significantly benefit from another round of revision. However, I won't object to accepting it if my co-reviewers champion it.

**Missing References:**

N/A

**Paper Topic And Main Contributions:**

This paper addresses the "representation degeneration problem" in cross-modal retrieval. To address this problem, the authors propose a novel method called INVGC, which establishes a graph topology structure within the dataset and applies graph convolution to effectively increase the distance between data points and separate representations. They also propose an advanced graph topology structure called LOCALADJ to improve the efficiency and effectiveness of INVGC. Experimental results show that both INVGC and INVGC w/LOCALADJ significantly alleviate the representation degeneration problem and improve retrieval performance.

**Questions For The Authors:**

See Weaknesses

**Reasons To Accept:**

A. The paper's overall structure is clear, the motivation described in the Introduction is easy to understand, and the previous work is summarized as well as possible in the Related Work.

B. The issues concerned in the article are important. The overall idea is clear and easy to understand.

C. Solid experiments and some theories support the paper, but some experimental details are not explained.

D. The proposed method is described in sufficient detail.

E. The ablation experiment was thoroughly studied.


**Reasons To Reject:**

A. The results were an improvement over the previous baseline but not much. In my opinion, there is an emphasis on comparing to the state of the art and demonstrating strong performance and it takes effort to find the place where differences to prior work and main factors for the performance differences are described.

B. The experimental scenarios can not simplify comparisons with other published works in the literature. I would have liked clear statements about how the results proposed here replicate or challenge previous experiments in other papers.

C. The range in which the parameters were searched is not given and it is not clearly written how the (validation?) data were used for model selection. How was the training loop stopped? How many epochs need to be trained on different datasets? Any early stopping?

D. The paper needs to give more details of the experiment to be convincing.


**Reproducibility:**

N/A: Doesn't apply, since the paper does not include empirical results.

**Reviewer Confidence:**

4: Quite sure. I tried to check the important points carefully. It's unlikely, though conceivable, that I missed something that should affect my ratings.

---

> ### Author Rebuttal · Authors · 2023-08-26
>
> We sincerely thank you for your time, efforts, and your detailed and positive comments. Below please find the responses (R) to some specific questions (Q).
>
> **Q(A) [Improvements are minor]:**
>
> >R: Different from finetuning or pretraining methods, our proposed $\textbf{InvGC}$ is a general **post-processing method** that does **not need any training or finetuning** on the model. It offers a ‘free’ retrieval improvement in performance without any training. From experimental results, we observe consistent improvements over base methods [1,2,3] after employing InvGC. Specifically, the improvement of R@1 of $\textbf{InvGC}$ w/$\textbf{LocalAdj}$ over CLIP is **2.77%** and **2.22%**, respectively for text-to-image and image-to-text retrieval, on MSCOCO.
>
> **Q(B) [Reproducing SOTAs (previous method)]:**
> >R: Thank you for your question!
> >1. As our proposed $\textbf{InvGC}$ is the **first post-processing** method addressing the data degeneration problem in the cross-modal retrieval, we design average pooling (i.e. $\textbf{AvgPool}$, Appendix C.5, Page 17) to serve as a baseline. Specifically, $\textbf{AvgPool}$ is the simplest variant of inverse graph convolution with binary adjacency matrix where the edge weight between a pair of nodes is 1 if they are neighbors, or 0 otherwise.
> >2. As for reproducing base methods, we use the public codes and default hyperparameters introduced in their papers. The details can be found in Table 10 (Page 16) in the appendix.
>
> **Q(C) [Parameters, training details]:**
> >R: Thank you for your comment! Our proposed $\textbf{InvGC}$ is a **post-processing method that does not require any training procedure**.
> > - $\textbf{Search space of hyperparameters}$: We set the search space of $r_g$ as [0,1] since it is used between the same modality, while for $r_q$, we use [0, 5]. We will report the hyperparameters selected in our experiments in the final version of our paper.
> > - $\textbf{Determining hyperparameters}$: We did use a validation set for selecting the hyperparameters, i.e., $r_g$ and $r_q$. For the dataset that does not have a validation set, we use a subset of the training set to determine the hyperparameters.
> > - $\textbf{Training details}$: As our proposed method $\textbf{InvGC}$ is a post-processing method, it does not have any training process. It runs on the embeddings generated by trained base models [1,2,3]. The details of training base models can be found in Table 10 (Page 16) in the appendix of our paper.
>
> **Q(D) [Details of the experiments]:**
> >R: Thank you for the suggestion! Due to the space limitation, the details of experiments are presented in Appendix C (Page15  - Page17), including benchmarks, methods, implementation of the proposed methods, and computational complexity. Moreover, to facilitate further research, we will release our code with detailed instructions upon publication.
>
> We sincerely thank you for your thoughtful comments and are open to further discussion on these matters.
>
> [1]Huaishao Luo, Lei Ji, Ming Zhong, Yang Chen, Wen Lei, Nan Duan, and Tianrui Li. 2022. Clip4clip: An empirical study of CLIP for end to end video clip retrieval and captioning. Neurocomputing, 508:293–304.
>
> [2] Valentin Gabeur, Chen Sun, Karteek Alahari, and Cordelia Schmid. 2020. Multi-modal transformer for video retrieval. In Computer Vision - ECCV 2020 - 16th European Conference, Glasgow, UK, August 23-28, 2020, Proceedings, Part IV, volume 12349 of Lecture Notes in Computer Science, pages 214–229. Springer.
>
> [3] Yiwei Ma, Guohai Xu, Xiaoshuai Sun, Ming Yan,Ji Zhang, and Rongrong Ji. 2022. X-CLIP: end-  to-end multi-grained contrastive learning for video- text retrieval. In MM ’22: The 30th ACM  International Conference on Multimedia, Lisboa,  Portugal, October 10 - 14, 2022, pages 638–647.  ACM.

---

### Meta-Review · Area_Chair_xSXi · 2023-09-19

**Recommendation:** 3

**Metareview:**

This research proposes a novel post-processing methodology to mitigate data degeneration in cross-modal retrieval.

Pros:
*The research methodology is robust and clear.
*The results and experiments are well-supported by ablation studies.

Cons:
* The feasibility of the setting requires additional elaboration.
* The mathematical notation is difficult to follow

---

### Decision · Program_Chairs · 2023-10-07

**Decision:**

Accept-Findings

**Comment:**

This research proposes a novel post-processing methodology to mitigate data degeneration in cross-modal retrieval.

Pros:
*The research methodology is robust and clear.
*The results and experiments are well-supported by ablation studies.

Cons:
* The feasibility of the setting requires additional elaboration.
* The mathematical notation is difficult to follow